# 7,8-Dihydroxyflavone modulates bone formation and resorption and ameliorates ovariectomy-induced osteoporosis

Fan Xue[1], Zhenlei Zhao[1], Yanpei Gu[1], Jianxin Han[1], Keqiang Ye[2]*, Ying Zhang[1]*

[1]Department of Food Science and Nutrition, College of Biosystems Engineering and Food Science, Zhejiang Key Laboratory for Agro-Food Processing; Zhejiang Engineering Center for Food Technology and Equipment, Zhejiang University, Hangzhou, China; [2]Department of Pathology and Laboratory Medicine, Emory University School of Medicine, Atlanta, United States

**Abstract** Imbalances in bone formation and resorption cause osteoporosis. Mounting evidence supports that brain-derived neurotrophic factor (BDNF) implicates in this process. 7,8-Dihydroxyflavone (7,8-DHF), a plant-derived small molecular TrkB agonist, mimics the functions of BDNF. We show that both BDNF and 7,8-DHF promoted the proliferation, osteogenic differentiation, and mineralization of MC3T3-E1 cells. These effects might be attributed to the activation of the Wnt/β-catenin signaling pathway as the expression of cyclin D1, phosphorylated-glycogen synthase kinase-3β (p-GSK3β), β-catenin, Runx2, Osterix, and osteoprotegerin (OPG) was all significantly up-regulated. Knockdown of β-catenin restrained the up-regulation of Runx2 and Osterix stimulated by 7,8-DHF. In particular, blocking TrkB by its specific inhibitor K252a suppressed 7,8-DHF-induced osteoblastic proliferation, differentiation, and expression of osteoblastogenic genes. Moreover, BDNF and 7,8-DHF repressed osteoclastic differentiation of RAW264.7 cells. The transcription factor c-fos and osteoclastic genes such as tartrate-resistant acid phosphatase (TRAP), matrix metalloprotein-9 (MMP-9), Adamts5 were inhibited by 7,8-DHF. More importantly, 7,8-DHF attenuated bone loss, improved trabecular microarchitecture, tibial biomechanical properties, and bone biochemical indexes in an ovariectomy (OVX) rat model. The current work highlights the dual regulatory effects that 7,8-DHF exerts on bone remodeling.

**\*For correspondence:**
kye@emory.edu (KY);
yzhang@zju.edu.cn (YZ)

**Competing interests:** The authors declare that no competing interests exist.

## Introduction

Bone is a dynamic and continuously renewed tissue that undergoes significant modification throughout the life cycle. Bone remodeling integrates osteoblast-mediated bone formation and osteoclast-mediated bone resorption as well as osteocytes within the bone matrix and osteoblast-derived lining cells that cover the surface of bone (*Crockett et al., 2011*; *Sims and Gooi, 2008*). Disrupting the balance between bone resorption and formation leads to bone disorders such as osteoporosis, multiple myeloma, and osteopetrosis (*Chen et al., 2018b*; *Feng and McDonald, 2011*; *Langdahl et al., 2016*; *Walsh and Gravallese, 2010*). Osteoporosis is defined by the World Health Organization (WHO) as a 'progressive systemic skeletal disease characterized by low bone mass and microarchitectural deterioration of bone tissue, with a consequent increase in bone fragility and susceptibility to fracture' (*Liu et al., 2019*). It is estimated that osteoporosis has affected hundreds of millions of people worldwide, predominantly postmenopausal women (*Brown, 2017*; *Compston et al., 2019*). Pharmacological agents for the treatment of osteoporosis can be classified as either anabolic or anti-resorptive, in the other words, either increasing bone building or blocking bone degradation (*Compston et al., 2019*; *Liu et al., 2016a*). However, treatment with anabolic agents such as parathyroid hormone (PTH) might lead to a simultaneous increase in bone resorption. For the

antiresorptive agent bisphosphonates, a concomitant decrease in bone formation can be observed (*Khosla and Hofbauer, 2017*; *Liu et al., 2018*). These drawbacks of the current therapies might be attributed to one target for these drugs that fail to couple bone formation and resorption. Researchers are constantly striving to explore dual-action agents that modulate bone remodeling for the treatment of osteoporosis (*Deal, 2009*).

A large number of factors have been implicated in regulating bone formation, including Wnt/β-catenin signaling pathway, which is crucial for osteoblast proliferation, differentiation, and survival (*Krishnan, 2006*). Activation of the canonical Wnt/β-catenin signaling pathway involves recruitment of a degrading complex, stabilization of β-catenin, regulation of transcription factors such as Runx2 and Osterix, and activation of Wnt target genes (*Baron and Kneissel, 2013*; *Karner and Long, 2017*; *Takada et al., 2009*). This pathway is active in osteoprogenitor cells or preosteoblasts such as MC3T3-E1 cells, and many signaling molecules in this pathway are developed as drug targets (*Bhukhai et al., 2012*; *Zhou et al., 2020*). Osteoclasts, formed by the differentiation and fusion of hematopoietic mononuclear precursor cells under the stimulation of the receptor activator of nuclear factor-κB ligand (RANKL), are the only cells responsible for bone resorption. After binding to receptor RANK, RANKL promotes osteoclastogenesis by activating several signaling pathways, including nuclear factor of activated T cells, cytoplasmic 1 (NFATc1)/c-fos pathways (*Kim and Kim, 2016*; *Zhao et al., 2017*). Furthermore, several genes such as cathepsin K, matrix metalloprotein-9 (MMP-9), and tartrate-resistant acid phosphatase (TRAP) that governing osteoclast differentiation and function are triggered after RANKL/RANK signal transduction cascades (*Boyle et al., 2003*; *Jiang et al., 2019*).

Brain-derived neurotrophic factor (BDNF) is a member of the neurotrophin family that regulates a variety of biological processes predominantly via binding to the transmembrane receptor tyrosine kinase B (TrkB). Nonetheless, BDNF-elicited TrkB signals are transient with a peak signal at 10 min and decreases at 60 min (*Liu et al., 2016a*; *Liu et al., 2014*). A systemic application study of BDNF found that BDNF had a short half-life and did not readily cross the blood-brain barrier (*Felts et al., 2002*). The outcomes of several clinical trials using recombinant BDNF are also disappointing, possibly because of the short in vivo half-life and poor delivery of BDNF (*Ochs et al., 2000*; *Thoenen and Sendtner, 2002*). These drawbacks impose an insurmountable hurdle to clinical application of BDNF. 7,8-Dihydroxyflavone (7,8-DHF), a plant-derived flavonoid, has been reported to overcome the above limitations of BDNF and identified as a functional BDNF mimic, which is able to induce TrkB dimerization and activate its downstream signaling molecules (*Jang et al., 2010*). 7,8-DHF is now extensively validated in various biochemical and cellular systems. It has been shown that BDNF could promote osteoblast differentiation, migration, and fracture healing (*Kilian et al., 2014*; *Liu et al., 2018*; *Zhang et al., 2020*; *Zhang et al., 2017*). Recently, we found that 7,8-DHF at the dose of 10 mg/kg·BW resulted in a marked increase in bone mineral density (BMD) irrespective of dietary conditions in female mice (*Zhao et al., 2021*). Inspired by the studies above, we wonder if 7,8-DHF could serve as a regulator on bone remodeling and exert antiosteoporosis effects. What's more, it is of interest to explore whether TrkB mediates the interaction between 7,8-DHF-induced osteoblastogenesis and Wnt/β-catenin signaling. In this study, we attempted to investigate the effects of BDNF and 7,8-DHF on osteoblastogenesis in osteoprogenitor MC3T3-E1 cells and on RANKL-induced osteoclastogenesis in RAW264.7 cells in vitro, particularly to elucidate the molecular mechanisms underlying functions of 7,8-DHF. In addition, the in vivo effects of 7,8-DHF on bone remodeling were examined in an ovariectomy (OVX)-induced osteoporosis rat model.

## Results

### 7,8-DHF promoted proliferation and differentiation of MC3T3-E1 osteoprogenitor cells

As shown in *Figure 1A*, both BDNF and 7,8-DHF displayed a positive effect on MC3T3-E1 cells by promoting cell growth ($p < 0.05$), except the lowest concentration of BDNF (25 ng/mL). To clarify whether the cell proliferation could be attributed to cell cycle alteration, cell nuclei were stained with propidium iodide (PI) and the cell cycle was analyzed using flow cytometry (*Figure 1B*). After treated with 7,8-DHF for 24 hr, the percentage of cells in S phase increased compared to control group ($p < 0.05$), with concomitant reduction of percentage of cells in the G0/G1 phase ($p < 0.01$). 7,8-

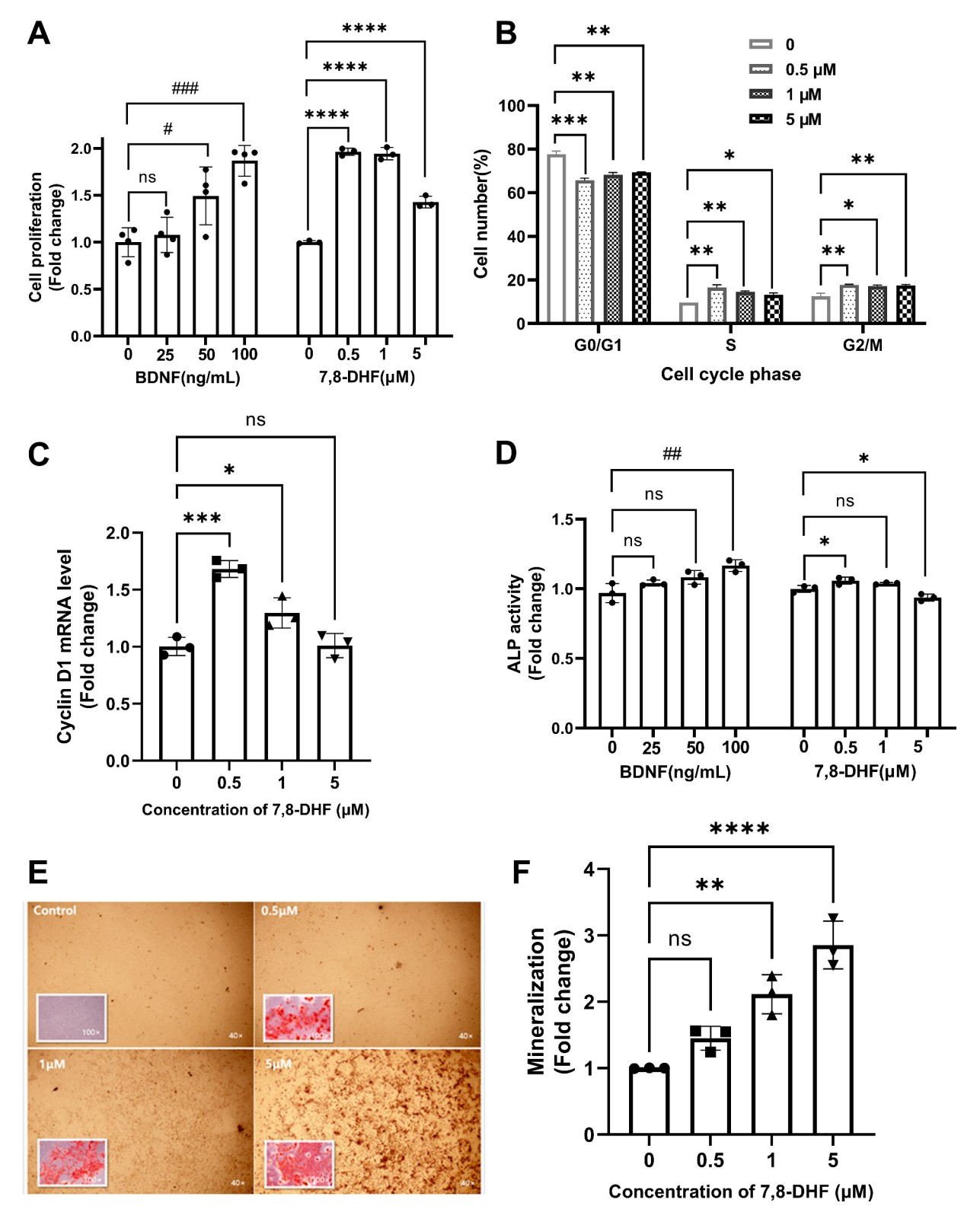

**Figure 1.** 7,8-Dihydroxyflavone (7,8-DHF) promoted proliferation and differentiation of MC3T3-E1 osteoprogenitor cells. (A) The effect of brain-derived neurotrophic factor (BDNF) or 7,8-DHF on the proliferation of MC3T3-E1 cells after treatment for 48 hr. (B) The percentage of MC3T3-E1 cells in each cell cycle phase treated with 7,8-DHF for 24 hr. Source file of gate parameters and regions chosen in the Modfit LT software for flow cytometry modeling was available in *Figure 1—source data 1*. (C) The effect of 7,8-DHF on the mRNA expression level of cyclin D1 was detected by

*Figure 1 continued on next page*

*Figure 1 continued*

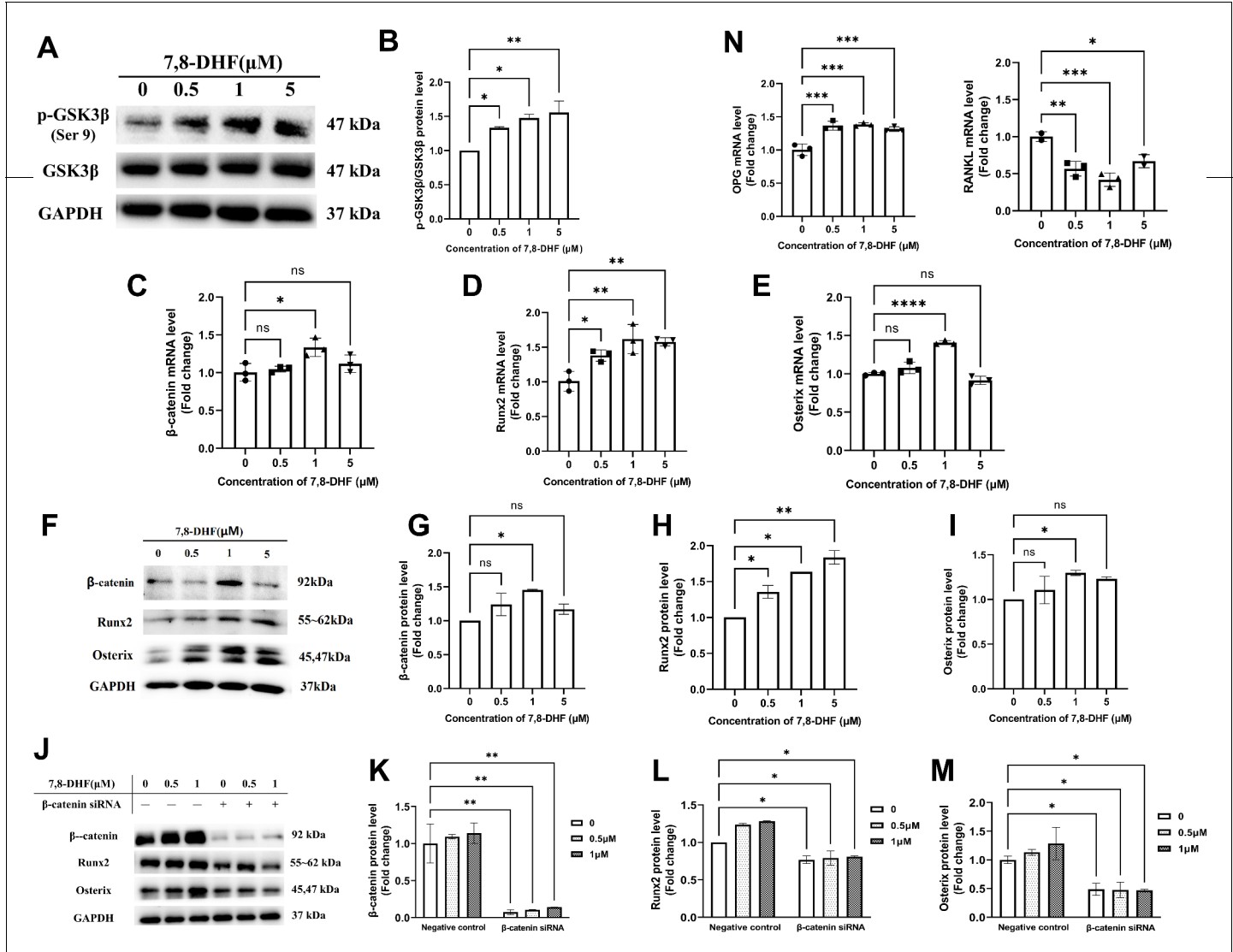

**Figure 2.** 7,8-Dihydroxyflavone (7,8-DHF) promoted osteogenesis via osteoblast-related signaling pathways. MC3T3-E1 cells were treated with or without 7,8-DHF for 3 days. The mRNA level was evaluated by quantitative real-time PCR (qRT-PCR) and the protein level was detected by western blot. GAPDH was used as an internal control. (**A**) The protein levels of p-GSK3β and GSK3β. (**B**) Quantification of the p-GSK3β band intensities normalized to total GSK3β band intensities in each case. (**C-E**) The mRNA levels of β-catenin, Runx2, and Osterix. (**F-I**) The protein levels of β-catenin, Runx2, and Osterix. The expression levels of target proteins in the 0 μM group were normalized to 1. (**J-K**) β-Catenin knockdown by siRNA was performed in MC3T3-E1 cells with or without 7,8-DHF treatment. The protein levels of β-catenin, Runx2, and Osterix. The expression levels of target proteins in the 0 μM of negative control group were normalized to 1. Representative images from three independent experiments are shown in (**A, F, J**). Source files of the full raw unedited blots and blots with the relevant bands labeled were provided in *Figure 2—source data 1*. (**N**) The mRNA levels of osteoprotegerin (OPG) and receptor activator of nuclear factor-κB ligand (RANKL). All results were expressed as mean ± SD (A-N: n = 3; *p < 0.05, **p < 0.01, ***p < 0.001, ****p < 0.0001, ns: not significant; A-I, N: one-way analysis of variance [ANOVA]; J-M: two-way ANOVA).

The online version of this article includes the following source data and figure supplement(s) for figure 2:

**Source data 1.** The original files of the full raw unedited blots and blots with the relevant bands labeled in *Figure 2A, F and J*.

**Figure supplement 1.** β-Catenin knockdown by siRNA was performed in MC3T3-E1 cells with or without 7,8-dihydroxyflavone (7,8-DHF) treatment.

**Figure supplement 2.** 7,8-Dihydroxyflavone (7,8-DHF) had no obvious influence on Smad2 and TAK1.

**Figure supplement 2—source data 1.** The original files of the full raw unedited blots and blots with the relevant bands labeled in *Figure 2—figure supplement 2A*.

DHF also up-regulated the proportion of cells in G2/M phase within tested dose range (p<0.05). The mRNA level of cyclin D1, a cell cycle controlling protein, was examined. 7,8-DHF caused a notable increase in the cyclin D1 mRNA level in MC3T3-E1 cells at the doses of 0.5 µM (p<0.001) and 1 µM (p<0.05), consistent with the trend observed in cell proliferation and cell cycle. However, the differences between control and 5 µM group were non-significant (*Figure 1C*).

The alkaline phosphatase (ALP) activity, an indicator of early-stage osteoblastic differentiation, was measured to investigate the effect of 7,8-DHF on the osteoblastic differentiation of MC3T3-E1 cells (*Figure 1D*). Results showed that cells treated with 0.5 µM 7,8-DHF exhibited higher ALP activity than those in the control group (p<0.05), while the 5 µM group showed a suppression effect (p<0.05). BDNF also significantly increased ALP activity at the concentration of 100 ng/mL (p<0.01). Mineralization is the final step of osteoblast differentiation and is often used as a critical marker to characterize bone formation. The calcium in mature osteoblasts combined with alizarin red S and formed orange-red chelates of poor water solubility. Calcium deposits in the control group were barely visible, whereas the calcification parts in treated cells increased evidently as the concentration of BDNF or 7,8-DHF raised (*Figure 1—figure supplement 1*, *Figure 1E*). The quantitative analysis in *Figure 1F* indicated that 7,8-DHF markedly enhanced mineralized nodule formation in a dose-dependent pattern, and the calcium content was increased about three times at the concentration of 5 µM (p<0.001).

## 7,8-DHF promoted osteogenesis via osteoblast-related signaling pathways

As Wnt/β-catenin signaling pathway plays an important role in bone homeostasis, GSK3β, β-catenin, and two key transcription factors (Runx2 and Osterix) that regulate osteogenesis were examined. 7,8-DHF inactivated GSK3β by increasing its phosphorylation (p<0.05) (*Figure 2A and B*), thus disrupted the β-catenin destruction complex. Compared with the control, 1 µM 7,8-DHF significantly increased mRNA and protein levels of β-catenin, Runx2, and Osterix (p<0.05) (*Figure 2C–I*). The mRNA and protein levels of Runx2 (*Figure 2D and H*), as well as the phosphorylated-glycogen synthase kinase-3β (p-GSK3β)/GSK3β value (*Figure 2A and B*), increased with the elevated concentrations of 7,8-DHF. A β-catenin knockdown experiment was performed to verify if β-catenin is the major downstream effector for inducing the expression of Runx2 and Osterix. The expression of β-catenin, Runx2, and Osterix was markedly reduced by β-catenin knockdown, as compared to negative control, in both the absence and presence of 7,8-DHF (p<0.05) (*Figure 2J–M*, *Figure 2—figure supplement 1*). The activities of other Runx2 inducers such as Smad2 and TAK1 were also determined. Results showed that 7,8-DHF had no obvious influence on Smad2 and TAK1 (*Figure 2—figure supplement 2*). We further evaluated the mRNA levels of two genes, osteoprotegerin (OPG) and RANKL. As shown in *Figures 2N*, 7,8-DHF promoted OPG mRNA expression but inhibited RANKL mRNA expression (p<0.05), thereby decreasing the OPG/ RANKL ratio compared with control group.

## Inhibition of TrkB blocked 7,8-DHF-mediated osteoblastogenesis

TrkB is the specific target of 7,8-DHF. To determine whether TrkB was involved in 7,8-DHF-mediated osteogenesis, MC3T3-E1 cells were pretreated with a Trks inhibitor, K252a. As shown in *Figure 3A and B*, K252a significantly suppressed 7,8-DHF-induced proliferation and the ALP activity in MC3T3-E1 cells. Western blot showed that K252a exerted apparent inhibition on the phosphorylation of TrkB at the dose of 100 nM (*Figure 3C*). The p-GSK3β/GSK3β ratio and the expression of osteoblast-related genes, such as β-catenin, Runx2, and Osterix, were reduced conspicuously, when K252a was added prior to 7,8-DHF (*Figure 3D–G*). Interestingly, K252a alone could cause a decrease in the protein level of Osterix, while the β-catenin and Runx2 expression was not affected (*Figure 3E–G*).

## 7,8-DHF inhibited RANKL-induced osteoclastogenesis

RAW264.7 cells were induced to differentiate into osteoclasts by RANKL. We investigated whether BDNF or 7,8-DHF possesses any inhibitory activity against RANKL-induced osteoclastic differentiation in RAW264.7 cells. TRAP is a well-known marker enzyme of osteoclastogenesis, so a TRAP staining was preliminarily observed. As shown in *Figure 4A and B*, the formation of TRAP-positive

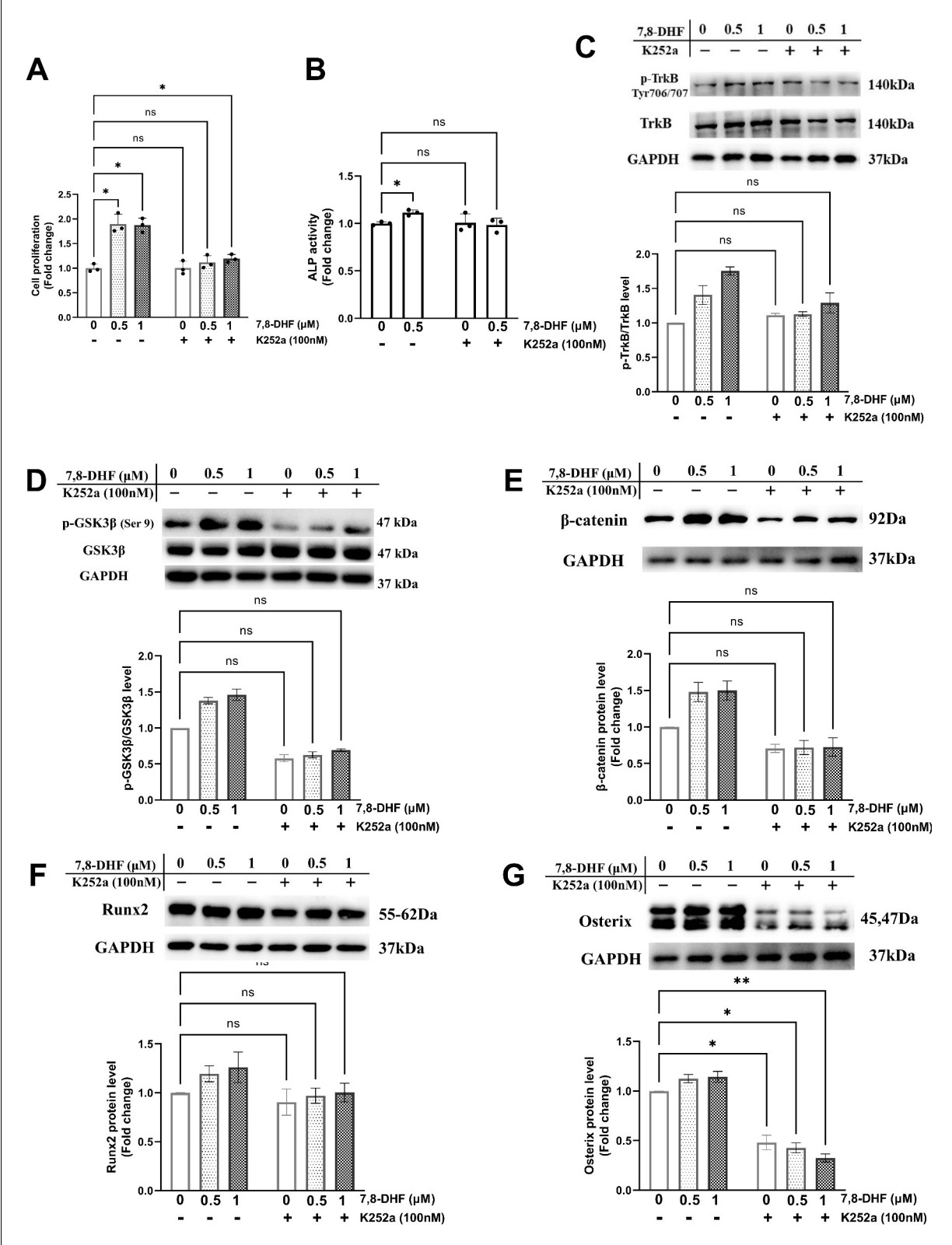

**Figure 3.** Chemical inhibition of TrkB blocked 7,8-dihydroxyflavone (7,8-DHF)-mediated osteogenesis. MC3T3-E1 cells were incubated with or without K252a (100 nM) for 1 hr followed by 7,8-DHF (0.5 μM or 1 μM). (**A**) K252a suppressed 7,8-DHF-induced proliferation of MC3T3-E1 cells after treatment for 48 hr. (**B**) K252a suppressed 7,8-DHF-elevated alkaline phosphatase (ALP) activity of MC3T3-E1 cells. (**C, D**) K252a inhibited 7,8-DHF-induced TrkB phosphorylation in MC3T3-E1 cells. Representative images from three independent experiments are shown in (**C**). (**E-G**) K252a inhibited 7,8-DHF-

*Figure 3 continued on next page*

*Figure 3 continued*

induced activation of Wnt/β-catenin signaling pathway in MC3T3-E1 cells. Representative images from three independent experiments are shown. The expression levels of target proteins in the control group (0 μM 7,8-DHF, without K252a) were normalized to 1. Source files of the full raw unedited blots and blots with the relevant bands labeled were provided in *Figure 3—source data 1*. All results were expressed as mean ± SD (A-G: n = 3, *p < 0.05, **p < 0.01, ns: not significant, two-way analysis of variance [ANOVA]).

The online version of this article includes the following source data for figure 3:

**Source data 1.** The original files of the full raw unedited blots and blots with the relevant bands labeled in *Figure 3C–G*.

multinucleated cells induced by RANKL was sharply reduced by BDNF or 7,8-DHF treatment (p<0.001), though 5 μM 7,8-DHF did not show any obvious effects. To further evaluate whether the inhibition of TRAP activity was caused by cytotoxic effects on RAW264.7 cells, cell survival tests based on the CCK-8 assay were performed. The result depicted in *Figure 4C* suggested that 7,8-DHF had no cytotoxicity on survival of RAW264.7 cells, even at the maximum concentration of 5 μM (p>0.05). Unlike 7,8-DHF, BDNF exhibited an accelerating effect on the growth of RAW264.7 cells (*Figure 4C*). Then, the change of mRNA level of osteoclast transcription factor c-fos was analyzed by the quantitative real-time PCR (qRT-PCR) assay; 0.5 and 1 μM 7,8-DHF significantly inhibited the expression of c-fos (p<0.05) (*Figure 4D*). On the contrary, 5 μM 7,8-DHF up-regulated the c-fos mRNA expression compared with the control (*Figure 4D*). Expression of resorption markers such as MMP-9 and Adamts5 was also examined. Results showed that 0.5 and 1 μM, but not 5 μM 7,8-DHF notably decreased the protein levels of both MMP-9 and Adamts5 (p<0.05) (*Figure 4F and G*). The influence of 7,8-DHF on the expression of these two markers was basically agreed with that reported in TRAP staining.

## 7,8-DHF alleviated osteoporosis phenotypes and enhanced biomechanical properties in OVX rat

As shown in *Figure 5A*, all rats displayed a similar initial body weight at the beginning of the study. The body weight of the OVX group was higher than the SHAM group 2 weeks after surgical operation. Treatment of 7,8-DHF prevented this distinct increase slightly, while both doses (OVX-L: 5 mg/kg/day; OVX-H: 10 mg/kg/day) of 7,8-DHF had no significant effect compared to the SHAM group. As expected, OVX caused serious atrophy of uterus, indicating the success of the surgical procedure. The uterine index decreased significantly in all ovariectomized rat compared to the SHAM group (p<0.001), and 7,8-DHF was devoid of any uterotrophic activity because uterine weight was not different among OVX, OVX-L, and OVX-H groups (*Figure 5B*). To compare the BMD among experimental groups, the left femur of rat was scanned for by dual-energy X-ray absorptiometry (DXA). OVX resulted in a significant decrease in the BMD that the OVX group was decreased to (0.146±0.090) g/cm$^2$ compared with the SHAM group (0.175±0.007) g/cm$^2$ (p<0.001). The OVX-H group showed (0.160±0.011) g/cm$^2$ of BMD, which was significantly greater than the OVX group and OVX-L group, though remains lower than the SHAM group (p<0.05) (*Figure 5C*).

The three-dimensional (3D) micro-CT images of the representative samples clearly displayed the differences of trabecular microstructure among the groups as represented in *Figure 5D*. High dose of 7,8-DHF (OVX-H: 10 mg/kg/day) revealed the potential to improve OVX-induced bone microstructure deterioration as the OVX-H rat rescued cancellous bone loss compared with OVX rat. A biomechanical testing apparatus was subsequently used to examine the effect of 7,8-DHF on the tibial mechanical properties of the experimental rat. The OVX rat showed a much worsening of the biomechanical properties, characterized by obviously decreased maximum load (p<0.05) (*Figure 5E*). Fortunately, this reduction was ameliorated in 7,8-DHF treatment groups and OVX-L rat exhibited a significantly higher maximum load than SHAM rat in particular (p<0.01). Interestingly, the results of the fracture deflection and the fracture strain among SHAM, OVX, and OVX-H groups had no significant difference, while both values of OVX-L group reached statistical significance compared to the other three groups (p<0.05) (*Figure 5F and G*).

## 7,8-DHF ameliorated the bone remodeling in OVX rat

As shown in *Table 1*, serum calcium (S-Ca) level was decreased in all ovariectomized rat and the measurement values did not show significant differences among the OVX, OVX-L, and OVX-H

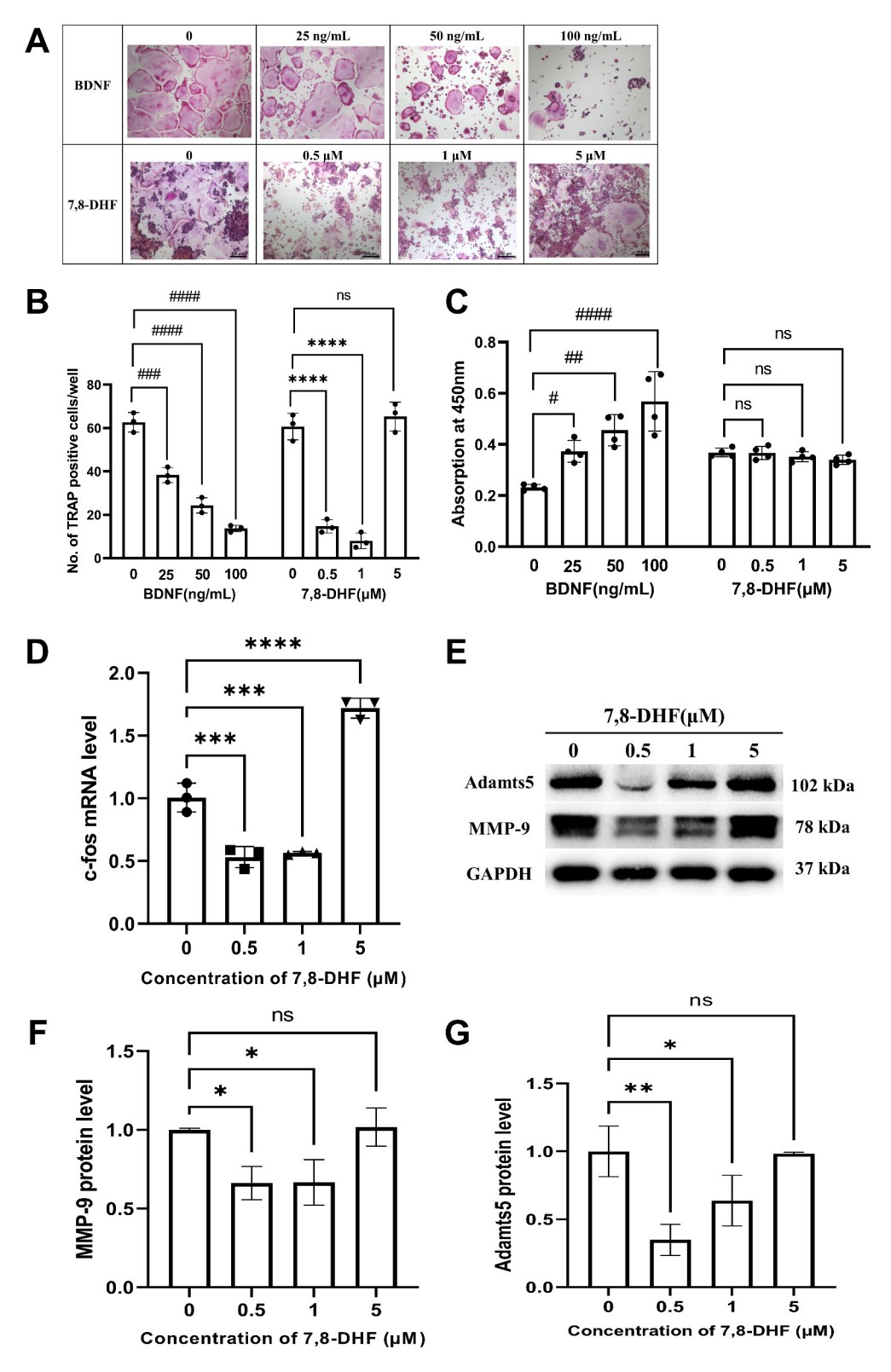

**Figure 4.** 7,8-Dihydroxyflavone (7,8-DHF) inhibited receptor activator of nuclear factor-κB ligand (RANKL)-induced osteoclastogenesis. (**A**) Representative images of tartrate-resistant acid phosphatase (TRAP)-positive multinucleated osteoclasts after the treatment with brain-derived neurotrophic factor (BDNF) or 7,8-DHF for 5 days (magnification: 100×, scale bar: 200 µm). Source files of micrographs used for the quantitative analysis are available in *Figure 4—source data 1*. (**B**) The average number of TRAP-positive multinucleated (nuclei ≥ 3) cells per cell. (**C**) The effects of BDNF or

*Figure 4 continued on next page*

*Figure 4 continued*

7,8-DHF on the cytoactive of RAW264.7 cells. (D) The mRNA level of c-fos. Results were normalized to the reference gene GAPDH. (E–G) The protein levels of matrix metalloprotein-9 (MMP-9) and Adamts5. GAPDH was used as an internal control. The expression levels of target proteins in the 0 µM group were normalized to 1. Representative images from three independent experiments are shown in (E). Source files of the full raw unedited blots and blots with the relevant bands labeled were provided in *Figure 4—source data 2*. All results were expressed as mean ± SD (B, D, F, G: n = 3, C: n=4; #p < 0.05, ##p < 0.01, ###p < 0.001, ####p < 0.0001, ns: not significant, BDNF-treated groups, one-way analysis of variance [ANOVA]; *p < 0.05, **p < 0.01, ***p < 0.001, ****p < 0.0001, ns: not significant, 7,8-DHF-treated groups, one-way ANOVA).

The online version of this article includes the following source data for figure 4:

**Source data 1.** Micrographs used for the quantitative analysis in tartrate-resistant acid phosphatase (TRAP) staining.

**Source data 2.** The original files of the full raw unedited blots and blots with the relevant bands labeled in *Figure 4E*.

---

groups. In contrast, urine calcium (U-Ca) was increased drastically in the OVX group (p<0.05), and 7,8-DHF could significantly reduce U-Ca level at both doses (OVX-L: 5 mg/kg/day; OVX-H: 10 mg/kg/day) to the SHAM group (p<0.05). A high circulating follicle-stimulating hormone (FSH) level occurs in response to ovarian failure. As expected, the FSH level in the OVX group was higher than that in the SHAM group. Both OVX-L and OVX-H groups exhibited reduced FSH and elevated estradiol (E2) levels compared to the OVX group, though did not show marked differences. In analysis of serum ALP and bone gal protein (BGP), two bone turnover markers, OVX resulted in notable up-regulation of ALP and BGP in the OVX group compared with the SHAM group (p<0.05). High concentration of 7,8-DHF (10 mg/kg/day) could significantly suppress the increase of ALP (p<0.05), whereas it was low concentration (5 mg/kg/day) that could eminently inhibit the increase with regard to BGP (p<0.05).

Histological analysis was conducted, including H&E staining and TRAP staining (*Figure 6A and B*). Left femur metaphysis in vertical plane with H&E staining in each group was shown in *Figure 6A*. Under the light microscope, serious bone loss was observed in the OVX group and the trabecular bone thickness was much thinner than that in other groups, accompanied with increased cavitas medullaris, which was subjected to fracture. In the OVX-L and OVX-H groups, the arrangement of trabecular bones tended to be more regular and the trabecular bone of the OVX-H group was much denser than those in the OVX group and close to the SHAM group. However, the OVX, OVX-L, and OVX-H groups all exhibited more lipid droplets in cavitas medullaris compared to the SHAM group, though the content of lipid droplets in the OVX-H group was slightly less than the OVX group. In addition, quantitative analysis of TRAP staining demonstrated that the number of multinucleated osteoclasts was increased remarkably in the OVX group compared with the SHAM group (p<0.05), and this sharp rise was remarkably decreased by 7,8-DHF treatment (p<0.01) (*Figure 6C*).

## Discussion

In this study, we performed a series of studies in vitro to demonstrate that 7,8-DHF promoted osteoblastic differentiation by interacting with TrkB receptors and subsequently reinforcing the Wnt/β-catenin pathway in MC3T3-E1 cells. In addition, 7,8-DHF inhibited RANKL-induced osteoclastogenesis by suppressing c-fos in RAW264.7 cells. Moreover, the effects of 7,8-DHF in vivo were explored through an OVX-induced bone-loss rat model. Results indicated that 5 mg/kg/day 7,8-DHF could improve the bone mass, microarchitecture, and bone strength of the OVX rat by inhibiting bone resorption and enhancing bone formation in vivo. Thus, our study demonstrates that 7,8-DHF could be an attractive therapeutic agent for osteoporosis treatment (*Figure 7*).

7,8-DHF has been identified as a promising small-molecular BDNF mimetic compound, which mimics the physiological actions of BDNF via directly binding to the extracellular domain of TrkB to trigger TrkB receptor dimerization and autophosphorylation (*Jang et al., 2010*). 7,8-DHF has been broadly validated in various BDNF-implicated disease models including a variety of neurological diseases, mental diseases, metabolic diseases, and obesity (*Agrawal et al., 2015*; *Bollen et al., 2012*; *Chen et al., 2019*; *Liu et al., 2016b*; *Pandey et al., 2020*; *Zhang et al., 2014*). Especially, 7,8-DHF displays robust therapeutic efficacy toward Alzheimer's disease (AD) (*Gao et al., 2016*; *Zhang et al., 2014*). At present, there are few related studies having been done on the physiological, pharmacological, and functional activities of 7,8-DHF on osteolytic disease, for example, osteoporosis. In fact,

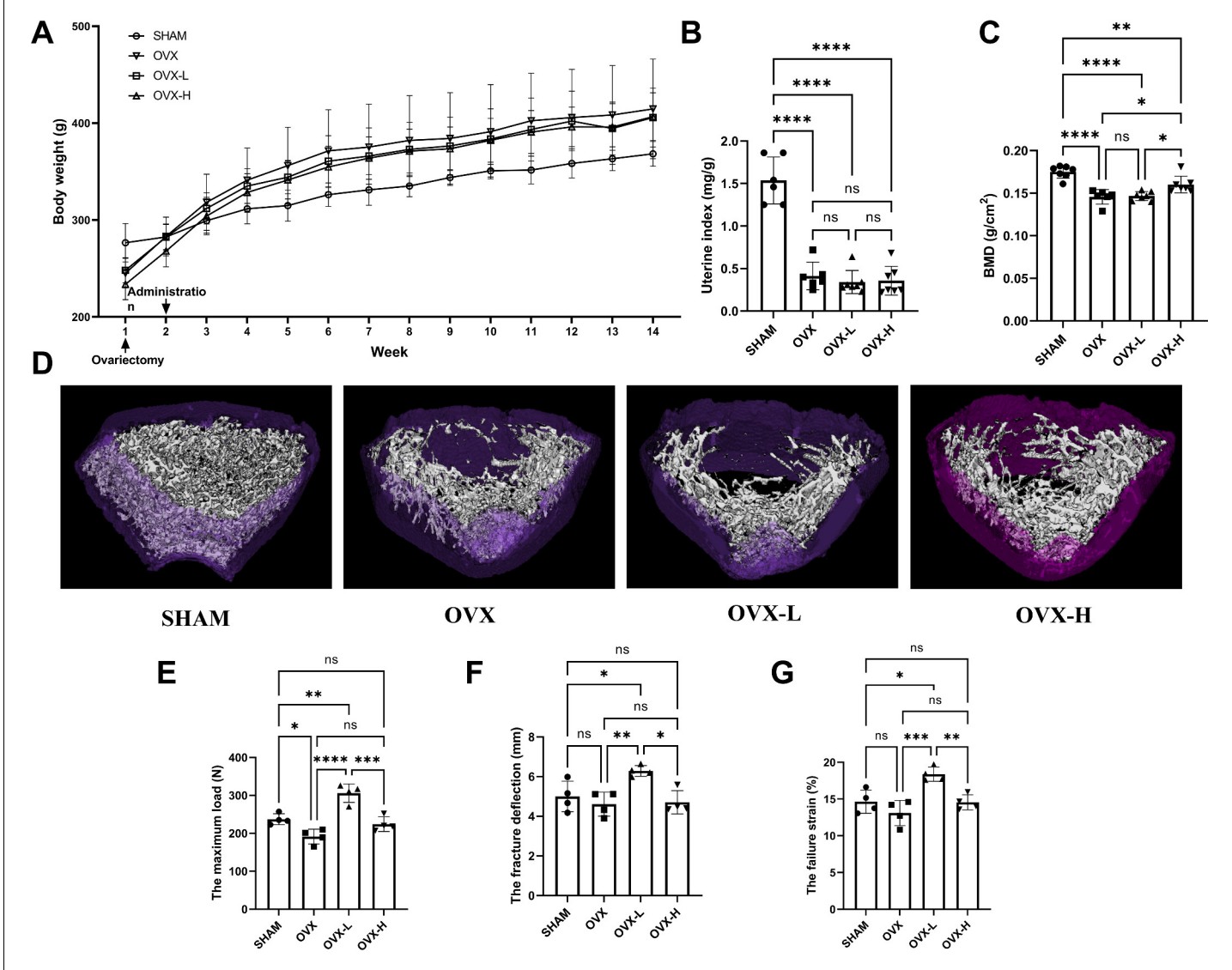

**Figure 5.** 7,8-Dihydroxyflavone (7,8-DHF) alleviated osteoporosis phenotypes and enhanced mechanical properties in ovariectomy (OVX) rats. (A) The body weights of all rats were recorded weekly during the experimental period. Source file of the weight record was available in *Figure 5—source data 1*. (B) Uteruses were isolated and weighed after euthanized. The uterine index was represented as uterus weight divided by body weight. Results were expressed as mean ± SD. (C) The bone mineral density (BMD) in left femur of rats by dual-energy X-ray absorptiometry (DXA). Results were expressed as mean ± SD (A-C: n = 6–7, *p < 0.05, **p < 0.01, ****p < 0.0001, ns: not significant, one-way analysis of variance [ANOVA]). (D) Representative micro-CT images from each group: three-dimensional (3D) architecture of trabecular bone within the distal metaphyseal femur region. Source files of the raw unedited images of proximal growth plate, trabecular structure, and cortical structure were available in *Figure 5—source data 2*. (E–G) Right femurs were isolated and subjected to a compression test for biomechanical property analysis. The maximum load (E), the fracture deflection (F) and the fracture strain (G) were evaluated for each group. Results were expressed as mean ± SD. All results were expressed as mean ± SD (n = 4, *p < 0.05, **p < 0.01, ***p < 0.001, ****p < 0.0001, ns: not significant, one-way ANOVA).

The online version of this article includes the following source data for figure 5:

**Source data 1.** Weight record of the experimental rat in each group.

**Source data 2.** Images of proximal growth plate, trabecular structure, and cortical structure.

recent studies reveal the important roles of BDNF in fracture healing. As early as in 2000, BDNF and TrkB were firstly found in bone-forming cells of mice, suggesting that they were involved in the regulation of bone formation as an autocrine and paracrine factor in vivo (*Asaumi et al., 2000*). In 2014, the detection of BDNF and TrkB in human fracture gap tissue during various stages of the bone

**Table 1.** The levels of bone biochemical indexes.

| Group | S-Ca (mmol/L) | U-Ca/Cr | FSH (mIU/mL) | E2 (pg/mL) | ALP (U/L) | BGP (ng/mL) |
|---|---|---|---|---|---|---|
| SHAM | 2.63±0.08[a] | 0.54±0.07[b] | 13.72±0.89 | 57.31±21.24 | 26.74±4.73[a] | 2.54±0.27[ab] |
| OVX | 2.49±0.08[b] | 1.29±0.49[a] | 14.24±0.70 | 50.64±6.45 | 47.03±10.49[c] | 2.48±0.22[a] |
| OVX-L | 2.42±0.04[b] | 0.42±0.19[b] | 13.98±1.59 | 58.28±19.49 | 43.58±4.07[bc] | 2.46±0.33[a] |
| OVX-H | 2.47±0.04[b] | 0.36±0.13[b] | 13.65±2.34 | 61.88±17.33 | 37.57±3.76[b] | 2.93±0.28[b] |

S-Ca: serum calcium; U-Ca/Cr: urine calcium/creatinine; FSH: follicle-stimulating hormone; E2: estradiol; ALP: alkaline phosphatase; BGP: bone gal protein. Results were presented as mean ± SD (n=6–7. Data with different letters in each group were significantly different at p<0.05).

formation process was reported (*Kilian et al., 2014*). Afterward, some researches confirmed that BDNF plays a positive role in accelerating bone union (*Kauschke et al., 2018a*; *Kauschke et al.,*

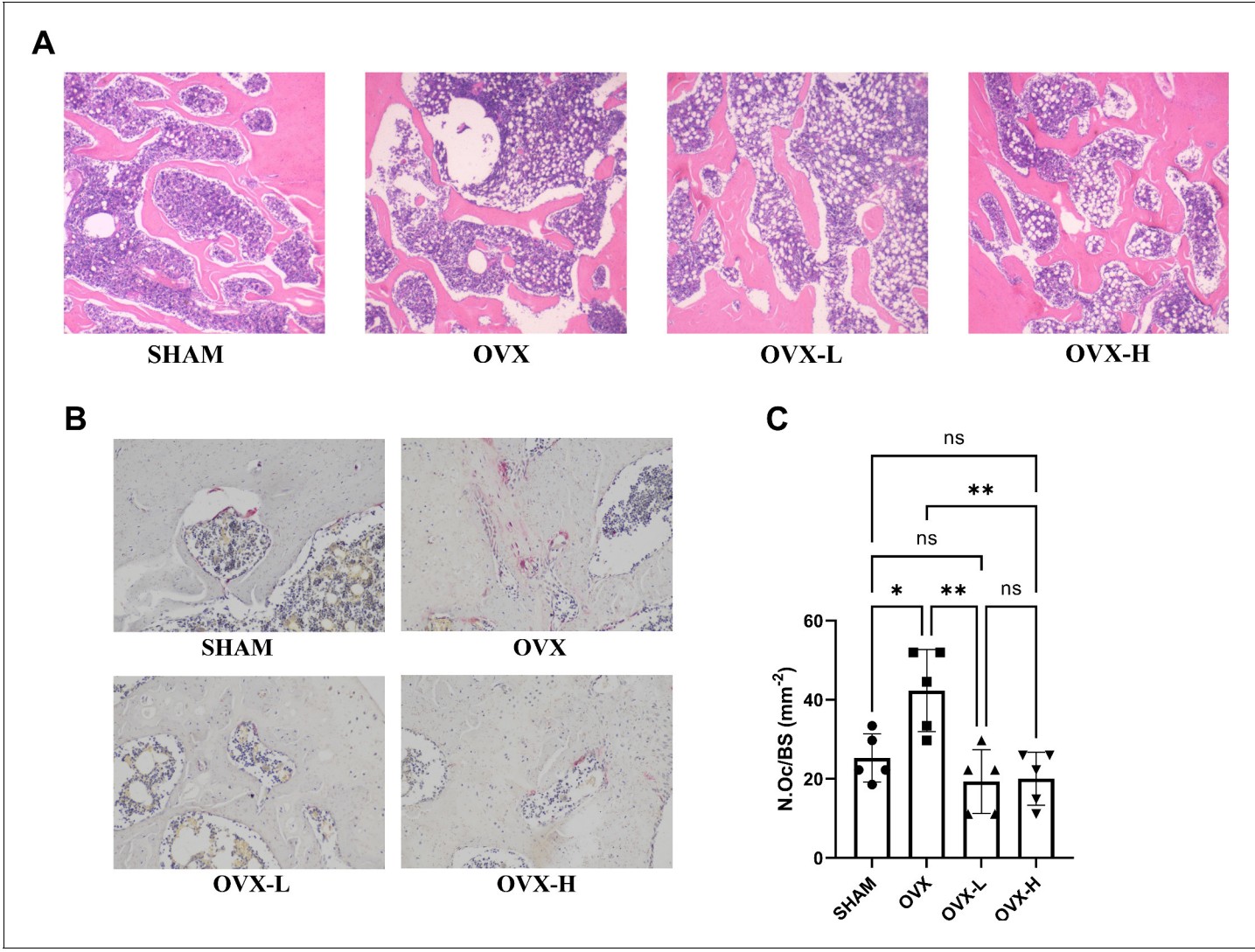

**Figure 6.** 7,8-Dihydroxyflavone (7,8-DHF) improved bone remodeling in ovariectomy (OVX) rat. (**A**) Representative images of left femurs sections stained with H&E. (**B**) Representative images of tartrate-resistant acid phosphatase (TRAP)-stained decalcified left femurs sections. Source files of micrographs used for the quantitative analysis are available in *Figure 6—source data 1*. (**C**) Quantitative statistics of osteoclast number per bone surface (N.Oc/BS). Results were expressed as mean ± SD (n = 5, *p < 0.05, **p < 0.01, ns: not significant, one-way analysis of variance [ANOVA]). The online version of this article includes the following source data for figure 6:

**Source data 1.** Micrographs used for the quantitative analysis in tartrate-resistant acid phosphatase (TRAP) staining.

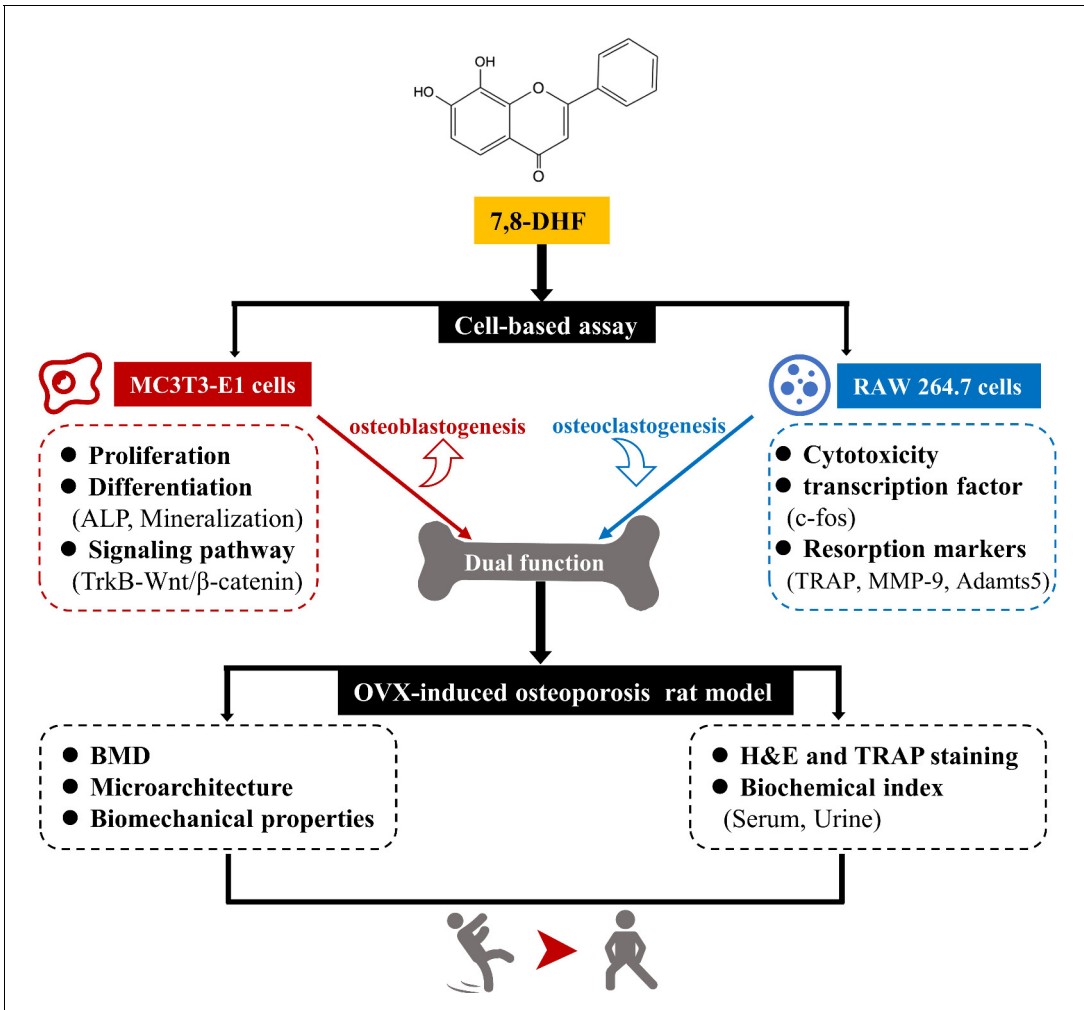

**Figure 7.** 7,8-Dihydroxyflavone (7,8-DHF) exerts dual regulation of bone remodeling. 7,8-DHF promotes osteoblastic proliferation and differentiation through the TrkB-Wnt/β-catenin signaling pathway and inhibits osteoclastogenesis at the same time in vitro. Furthermore, 7,8-DHF improves bone mass, trabecular microarchitecture, tibial biomechanical properties, and bone biochemical indexes in vivo as indicated in an ovariectomy (OVX)-induced osteoporosis rat model.

*2018b*; *Liu et al., 2018*). Recently, a study provided further understanding for the underlying mechanism of BDNF in fracture healing that BDNF contributed to MC3T3-E1 migration and increased the expression level of integrin β1 (*Zhang et al., 2020*). Considering the fact that 7,8-DHF mimics so many aspects of BDNF, we hypothesized that there may be a regulatory network involving 7,8-DHF and bone remodeling. To our knowledge, this is the first study reporting the therapeutic potential of 7,8-DHF toward osteoporosis.

Bone homeostasis requires the coordinated recruitment of osteoblasts and osteoclasts to sites of remodeling (*Phan et al., 2020*). The crosstalk among bone cells is maintained and orchestrated by a complex system, which includes hormones, growth factors, and physical activity (*Jiang et al., 2019*). Due to the tight coupling and synchronization of bone formation and resorption, the limitation of clinically used drugs is stimulation or inhibition of both phases at the same time. For example, bisphosphonates decrease bone degradation, but also reduce bone formation. What's worse, it may cause serious jaw and subtrochanteric fractures, which are mostly attributed to its effects that induce osteoclast cytotoxicity (*Gasser, 2019*; *Khosla and Hofbauer, 2017*; *Reid, 2011*). Likewise, drugs such as PTH or PTH-related protein mainly enhance the bone formation and meanwhile stimulate bone resorption (*Leder, 2017*). In our present study, 7,8-DHF, as well as BDNF, exhibited the ability to regulate bone remodeling by synchronously promoting osteoblastic differentiation and restraining

the formation of osteoclasts. Previous studies reported that BDNF is a potential osteoclastogenic factor in multiple myeloma, and myeloma-derived BDNF could promote RANKL secretion and osteoclasts formation (*Ai et al., 2012*; *Sun et al., 2012*). Nevertheless, in our study we found that BNDF at the concentration of 25 ng/mL markedly inhibited the formation of TRAP-positive multinucleated osteoclasts derived from RAW264.7 cells, though contributed to the growth of RAW264.7 cells. Of note, we observed that the TRAP-positive cells, c-fos mRNA level, the expression of MMP-9, and Adamts5 were decreased when 0.5 or 1 μM 7,8-DHF was used, while the inhibitory effect was lost at dose of 5 μM. This observation indicated that the effective therapeutic dose range of 7,8-DHF for suppression of bone resorption might be narrow. The underlying molecular mechanism accounting for this curve remains unclear. Nevertheless, the prodrug of 7,8-DHF, R13, also displayed the similar bell-shaped pharmacological curve in therapeutic treating AD (*Chen et al., 2018a*). Therefore, optimization of the chemical structure of 7,8-DHF or development of its analogues might exhibit a positive dose-dependent effect, extending the inhibitory activity for osteoclastogenesis.

Besides 7,8-DHF, other natural products or their derivatives such as emodin, wedelolactone, rhein, and psoralen have also showed dual effects on bone remodeling (*Jiang et al., 2019*; *Kim et al., 2014*; *Liu et al., 2016a*; *Newman and Cragg, 2020*; *Zhang et al., 2019*). One of the possible causes of their dual effects might result from that they can target multiple signaling pathways and display various beneficial pharmacological effects (*Jiang et al., 2019*). The canonical Wnt/β-catenin signaling pathway has been demonstrated to be responsible for a variety of biological processes, including the regulation of osteoblast proliferation and differentiation. β-Catenin is the principal mediator of Wnt/β-catenin signaling pathway (*Baron and Kneissel, 2013*; *Gordon and Nusse, 2006*). In the absence of extracellular Wnt ligands, β-catenin is recruited into a degrading complex comprising GSK-3β, the scaffold protein axin, and adenomatous polyposis coli (*Moon et al., 2004*). Thus, the content of the β-catenin protein in normal cells is generally kept very low. In the presence of Wnt ligands, on the other hand, GSK3β is phosphorylated to yield inactive p-GSK3β, and the degrading complex is depolymerized, which allows β-catenin to avoid the degradation by the proteasome and accumulate in the cytoplasm (*Chen et al., 2021*; *Wang et al., 2019*). The accumulated β-catenin translocates into the nucleus, binds with T-cell-specific transcription factor/lymphoid enhancer binding factor, and activates a series of target gene, such as cyclin D1, Runx2, and Osterix. It has been reported that BDNF is involved in the regulation of GSK3β activity both in the retina tissue in vivo and in the neuronal cells in vitro (*Gupta et al., 2014*). In this research, we observed that 7,8-DHF induced phosphorylation of GSK3β at the serine nine residue, thus promoting the aggregation and localization of β-catenin in the nucleus.

Cyclin D1, whose activity is required for G1/S transition, is one of the key cell cycle regulators. It is synthesized during G1 phase and assembles with either cyclin-dependent kinase 4 (CDK4) or CDK6 (*Fujita et al., 2002*; *Ozaki et al., 2007*). Treatment of MC3T3-E1 cells with 7,8-DHF could upregulate the mRNA level of cyclin D1, thus stimulating more cells at S phase but less at G0/G1 phase. These results indicated active DNA synthesis, which was consistent with cell proliferation results. Two transcription factors, Runx2 and Osterix/SP7, have previously been shown to be essential for the differentiation of osteoblasts (*Ryoo et al., 2006*; *Yang and Karsenty, 2002*). Runx2 not only acts as a scaffold for nucleic acids and regulatory factors involved in skeletal gene expression, but also promotes skeletal development on different levels, including differentiation of mesenchymal progenitors into osteoblasts, chondrocyte maturation, and skeletal morphogenesis (*Kokabu et al., 2016*; *McGee-Lawrence et al., 2014*). Osterix plays an important role at a later step in the process of osteoblast differentiation, that is, the differentiation of pre-osteoblasts into mature osteoblasts and osteocytes. Moreover, it is a key target of mechanical signals that affect bone biology (*Fan et al., 2007*). In the present study, we have shown that 7,8-DHF could enhance osteoblastic differentiation by stimulating Runx2 and Osterix via β-catenin. The possibility that 7,8-DHF enhanced the activity of other Runx2 inducers such as Smad2 and TAK1 was excluded.

What molecule(s) served as a 'bridge' between 7,8-DHF-induced osteogenesis and the activation of the Wnt/β-catenin pathway? It has been reported that 7,8-DHF inhibits obesity through activating muscular TrkB. The anti-obesity activity of 7,8-DHF is muscular TrkB-dependent and 7,8-DHF cannot mitigate diet-induced obesity in female muscle-specific TrkB-knockout mice (*Chan et al., 2015*). Besides, the promotion effect of BDNF on osteoblast migration and fracture healing is also via TrkB. The specific inhibitor for Trks, K252a, can suppress BDNF-induced cell migration, integrin β1 expression, and activation of ERK1/2 and AKT signaling pathways (*Zhang et al., 2020*). We thus

hypothesized that TrkB is the link between 7,8-DHF and Wnt/β-catenin reinforcement during osteogenic differentiation. As expected, 7,8-DHF-induced activation of Wnt/β-catenin signaling pathway in MC3T3-E1 cells was diminished by inhibiting TrkB with K252a. In the presence of K252a, the elevation of the expression of p-GSK3β, β-catenin, Runx2, and Osterix was reduced, suggesting that 7,8-DHF acts on TrkB to alter the expression of critical genes involved in the Wnt/β-catenin signaling pathway.

Differentiated osteoblasts subsequently produce positive and negative regulators of osteoclastogenesis, such as RANKL, a member of the tumor necrosis factor-ligand family, and its natural decoy receptor OPG, respectively (*Kong et al., 1999*; *Lacey et al., 1998*). RANKL binds to its receptor RANK on osteoclast progenitors and activates c-fos to induce transcription of genes related to osteoclasts (*Leibbrandt and Penninger, 2008*). Whereas OPG competitively binds to RANKL and prevents activation of RANK, leading to the inhibition of osteoclast recruitment. Otherwise, the production of OPG is tightly dependent on the Wnt/β-catenin signaling pathway (*Li et al., 2020*). From the experimental results, we infer that 7,8-DHF-induced increase of OPG might be involved in the activation of Wnt/β-catenin signaling pathway, and the decreased RANKL probably down-regulated the transcription of c-fos to a certain degree, leading to the suppression of NF-κB pathway and ultimately inhibiting the osteoclastic differentiation of RAW264.7 cells. The RANK/RANKL/OPG axes couple osteoblast and osteoclast activities, thereby controlling the balance between bone formation and resorption (*Boyle et al., 2003*). The ratio between OPG and RANKL is considered a quantitative osteogenic activity standard (*Liu et al., 2012*). The current study suggests that 7,8-DHF positively modulates OPG/RANKL balance, as OPG/RANKL ratio was increased in response to 7,8-DHF, representing the capability of 7,8-DHF to inhibit osteoclast activity.

Considering the fascinating dual functional effect of 7,8-DHF on osteoblasts and osteoclasts in vitro, we further proved that 7,8-DHF exerted a protective effect against osteoporosis in vivo using an OVX-induced osteoporotic rat model. In agreement with the previous observations, ovarian hormone deficiency, caused by surgical removal of ovaries, significantly increased body weight (*Zhao et al., 2019*; *Zhou et al., 2020*), but this excess body weight gain was partially prevented by 7,8-DHF. Besides, OVX can accelerate bone remodeling, an uncoupling between bone resorption and formation (*Libouban et al., 2003*). As the resorption phase is short but the period required for osteoblastic replacement is long, any increase in the rate of bone remodeling will result in a loss of bone mass (*Raisz, 2005*). It is necessary to evaluate the true impact of a treatment on the quality of trabecular bone because trabecular bone is more readily lost due to OVX in this animal model (*Maimoun et al., 2012*). Radiographic and histomorphology analysis showed that 7,8-DHF attenuated bone loss and alleviated the compromised trabecular microstructure in ovariectomized rat. However, neither of the doses was able to restore trabecular bone completely. The loss of bone mass is usually accompanied by enhanced levels of the bone turnover markers, such as ALP and BGP (*Park et al., 2011*). The relief of the high bone turnover status in the 7,8-DHF-treated groups was evidenced by lower serum levels of ALP and BGP. Moreover, an increase in urinary calcium excretions might contribute to the reduction of BMD (*Park et al., 2011*). Results showed that 7,8-DHF prevented the OVX-induced increase in urinary Ca excretion. Although such inhibition generally would be considered beneficial, biomechanical properties of bone may be decreased if bone remodeling is inhibited excessively. Fortunately, we found that 7,8-DHF at a lower dose (5 mg/kg/day) significantly improved the biomechanical parameters in OVX rats.

In summary, the present study evaluates the potential of 7,8-DHF as a novel dual regulator for bone formation and bone resorption. We found that 7,8-DHF promoted osteoblastic proliferation and differentiation through the TrkB-Wnt/β-catenin signaling pathway and inhibited osteoclastogenesis at the same time. Furthermore, 7,8-DHF could alleviate osteoporosis phenotypes, enhance mechanical properties, and ameliorate bone remodeling in OVX rat. Notably, 7,8-DHF is orally bioactive and is safe for chronic treatment. Hence, it acts as a good compound for further medicinal application to maximize its therapeutic properties toward various bone disorders, especially postmenopausal osteoporosis.

## Materials and methods

### Key resources table

| Reagent type (species) or resource | Designation | Source or reference | Identifiers | Additional information |
|---|---|---|---|---|
| Cell line (*Mus musculus*) | MC3T3-E1 Subclone 14 | The Cell Bank of the Chinese Academy of Sciences | Cat #:GNM15 RRID:CVCL_5437 | |
| Cell line (*Mus musculus*) | RAW 264.7 | The Cell Bank of the Chinese Academy of Sciences | Cat #:SCSP-5036 RRID:CVCL_0493 | |
| Peptide, recombinant protein | BDNF | PeproTech | Cat #:450–02 | |
| Peptide, recombinant protein | RANKL | R and D Systems | Cat #:462-TEC-010 | |
| Chemical compound, drug | 7,8- Dihydroxyflavone; 7,8-DHF | Tokyo Chemical Industry | Cat #:D1916 | |
| Chemical compound, drug | Ascorbic acid | Aladdin | Cat #:A103539 | |
| Chemical compound, drug | β-Glycerophosphate disodium salt hydrate | Sigma-Aldrich | Cat #:G9422 | |
| Chemical compound, drug | Dexamethasone | Sigma-Aldrich | Cat #:D4902 | |
| Chemical compound, drug | K252a | Cell Signaling | Cat #:12754 | |
| Antibody | Rabbit monoclonal anti-GSK3β | Huabio | Cat #:ET1607-71 | WB: 1:1000 |
| Antibody | Rabbit monoclonal anti-p-GSK3β (Ser 9) | Huabio | Cat #:ET1607-60 | WB: 1:1000 |
| Antibody | Rabbit monoclonal anti-β-catenin | Abcam | Cat # ab32572 | WB: 1:5000 |
| Antibody | Rabbit monoclonal anti-Osterix | Abcam | Cat #:ab209484 | WB: 1:1000 |
| Antibody | Rabbit monoclonal anti-Runx2 | Cell Signaling | Cat #:12556 | WB: 1:1000 |
| Antibody | Rabbit monoclonal anti-Smad2 | Huabio | Cat #:ET1604-22 | WB: 1:1000 |
| Antibody | Rabbit monoclonal anti-p-Smad2 (S250) | Huabio | Cat #:ET1612-32 | WB: 1:1000 |
| Antibody | Rabbit monoclonal anti-TAK1 | Huabio | Cat #:ET1705-14 | WB: 1:1000 |
| Antibody | Rabbit polyclonal anti-TrkB | Affinity | Cat #:AF6461 | WB: 1:1000 |
| Antibody | Rabbit polyclonal anti-P-TrkB (Tyr706) | Affinity | Cat #:AF3461 | WB: 1:1000 |
| Antibody | Rabbit monoclonal anti-MMP-9 | Abcam | Cat #:ab228402 | WB: 1:1000 |
| Antibody | Rabbit polyclonal anti-Adamts5 | Huabio | Cat #:1903–32 | WB: 1:1000 |
| Antibody | Rabbit polyclonal anti-GAPDH | Proteintech | Cat #:10494–1-AP | WB: 1:5000 |
| Sequence-based reagent | qRT-PCR primers | This paper | | See *Supplementary file 1* |
| Sequence-based reagent | siRNA | This paper | | See Materials and methods, Small interfering RNA transfection section |

*Continued on next page*

*Continued*

| Reagent type (species) or resource | Designation | Source or reference | Identifiers | Additional information |
|---|---|---|---|---|
| Commercial assay or kit | Cell Counting Kit-8; CCK-8 | Biosharp | Cat #:BS350A | |
| Commercial assay or kit | DNA Content Quantitation Assay (Cell Cycle) kit | Solarbio | Cat #:CA1510 | |
| Commercial assay or kit | BCA Protein Assay Kit | Solarbio | Cat #:PC0020 | |
| Commercial assay or kit | ALP assay kit | Nanjing Jiancheng | Cat #:A059-2 | |
| Commercial assay or kit | Alizarin Red S solution (1%, pH 4.2) | Solarbio | Cat #:G1452 | |
| Commercial assay or kit | Acid phosphatase, leukocyte (TRAP) kit | Sigma-Aldrich | Cat #:387A | |
| Commercial assay or kit | TRIzol Reagent | Invitrogen | Cat #:15596026 | |
| Commercial assay or kit | PrimeScript RT reagent Kit with gDNA Eraser | TaKara | Cat #:RR047A | |
| Commercial assay or kit | PowerUp SYBR Green Master Mix | Applied Biosystems | Cat #:A25742 | |
| Commercial assay or kit | Lipofectamine 3000 | Invitrogen | Cat #:L3000001 | |
| Commercial assay or kit | Rat FSH ELISA kit | Cusabio | Cat #:CSB-E06869r | |
| Commercial assay or kit | Rat E2 ELISA kit | Cusabio | Cat #:CSB-E05110r | |
| Commercial assay or kit | Rat BGP ELISA kit | Cusabio | Cat #:CSB-E05129r | |
| Software, algorithm | ModFit LT | Verity Software House (http://www.vsh.com/products/mflt/index.asp) | RRID:SCR_016106 | |
| Software, algorithm | ImageJ | ImageJ (http://imagej.nih.gov/ij/) | RRID:SCR_003070 | |
| Software, algorithm | GraphPad Prism | GraphPad (https://www.graphpad-prism.cn/) | RRID:SCR_002798 | Version 9 |

## Cell culture and differentiation

The osteoblastic cell line MC3T3-E1 Subclone 14 and the monocyte/macrophage-like cell line RAW264.7 cells were purchased from the Cell Bank of the Chinese Academy of Sciences (Shanghai, China). Identities of cell lines were authenticated by the vendors using STR profiling. They confirmed that the cell lines tested negative for mycoplasma contamination. MC3T3-E1 cells were cultured in α-MEM supplemented with 10% FBS and 1% penicillin-streptomycin in a humidified incubator under 5% $CO_2$ balanced air at 37°C. To induce differentiation and/or mineralization, cells were seeded into 6- or 12-well plates and cultured in osteogenic differentiation medium (α-MEM containing 50 μg/mL ascorbic acid, 10 mM β-glycerophosphate, and 10 nM dexamethasone). The culture medium was changed every 2 days. RAW264.7 cells were maintained in DMEM supplemented with 10% FBS and 1% penicillin-streptomycin at 37°C with 5% $CO_2$ in a humidified atmosphere. For the induction of osteoclast differentiation, RAW264.7 cells were cultured in the presence of 50 ng/mL RANKL, and the medium was displaced with fresh differentiation medium every 2 days.

## CCK-8 assay

The proliferation of MC3T3-E1 cells and cytotoxic effect of BDNF or 7,8-DHF on RAW264.7 cells were determined by CCK-8 assay according to the manufacturer's instructions. Cells were seeded in

96-well plates. BDNF was initially reconstituted in water to 0.1 mg/mL and then diluted in a buffer containing 0.1% bovine serum albumin for extended storage. 7,8-DHF was dissolved in DMSO at 100 mM as a stock solution and diluted with medium. A final DMSO concentration in the culture was less than 0.01% and did not show observably artificial or cytotoxic effects. After 24 hr, the medium was replaced with fresh medium containing different concentrations of BDNF or 7,8-DHF. Cells were cultivated for definite time, followed by CCK-8 solution for 4 hr at 37℃. The absorbance of the resulting solution at 450 nm was measured via a Microplate Reader (Eon, BioTek, Winooski, VT).

## Cell cycle analysis

The distribution of MC3T3-E1 cells in the cell cycle phases was analyzed by measuring DNA content using a flow cytometer (FACSCanto, BD Bioscience, San Jose, CA). MC3T3-E1 cells were seeded in six-well plates and treated with different concentrations of 7,8-DHF for 24 hr. After that, cells were harvested with trypsin and the cell pellets were fixed in ice-cold 70% ethanol at 4℃ overnight. Then the fixed cells were recovered by centrifugation, rewashed with PBS, incubated with 100 μL RNase at 37℃ for 30 min, and stained with 400 μL PI at 4℃ for 30 min in the dark prior to the measurement.

## ALP activity assay

ALP activity in MC3T3-E1 cells was determined by a colorimetric assay using disodium phenyl phosphate as the substrate. MC3T3-E1 cells cultured in six-well plates were treated with different concentrations of BDNF or 7,8-DHF for 7 days. After treated with BDNF or 7,8-DHF for 7 days, cells were lysed with RIPA (radio immunoprecipitation assay) buffer. The protein concentration of the supernatant was determined by a BCA protein assay. The ALP activity was measured by an ALP assay kit (Nanjing Jiancheng Bioengineering, China) according to the protocols provided by the manufacturers and normalized to the protein concentration.

## Mineralization assay

The degree of mineralization was determined by alizarin red S staining in 12-well plates. MC3T3-E1 cells were incubated with different concentrations of BDNF or 7,8-DHF for 21 days. MC3T3-E1 cells were fixed with 4% paraformaldehyde for 15 min and rinsed with ddH$_2$O. The cells were then stained with 1% alizarin red S (pH 4.2) for 30 min at room temperature. The samples were observed under light microscope, and the representative pictures were photographed to detect the bone nodules (calcium precipitates) formation. To quantify the degree of mineralization, the bound stain was eluted with 10% (w/v) cetylpyridinium chloride and the absorbance of the solubilized dye was recorded at 561 nm.

## TRAP staining

TRAP histochemical staining of RAW264.7 cells was performed using an acid phosphatase, leukocyte (TRAP) kit (Sigma-Aldrich, St. Louis, MO). RAW264.7 cells were cultured in medium containing different concentrations of 7,8-DHF in 48-well plates in the presence of RANKL (50 ng/mL). Following incubation for 5 days, cells were fixed with 4% paraformaldehyde for 20 min at room temperature. TRAP staining was performed according to the manufacturer's instructions. TRAP-positive multinucleated (more than three nuclei) cells were photographed under manual inverted microscope (Leica DMI3000 B).

## Quantitative real-time PCR

Cells were seeded in six-well plates and treated with different concentrations of 7,8-DHF. Total RNA was extracted with TRIzol Reagent (Thermo Fisher Scientific, Inc, Waltham, MA) and reverse-transcribed to cDNA using PrimeScript RT reagent Kit with gDNA Eraser (TaKaRa, Beijing, China) according to the manufacturer's instructions. The detailed sequences of the specific primers are shown in *Supplementary file 1*. cDNA was amplified using PowerUp SYBR Green Master Mix (Applied Biosystems, Foster City, CA). Each PCR reaction was carried out in a single 96-well PCR Microplates (Axygen, Union City, CA) using the QuantStudio3 Real-Time PCR System (Applied Biosystems, Foster City, CA). The expression levels of target genes were calculated using the $2^{-\Delta\Delta Ct}$ method and normalized to the reference gene GAPDH.

## Western blotting

Cell lysates were prepared with RIPA buffer containing fresh protease and phosphatase inhibitor PMSF (1 mM). Protein concentration was quantified using the BCA Protein Assay Kit. Equal amounts of proteins were run in an 8% or 10% polyacrylamide gel and transferred to polyvinylidene difluoride membranes. After 1 hr blocking with 5% (w/v) skim milk at room temperature, the membrane was incubated with indicated primary antibodies overnight at 4°C. Horseradish peroxidase-conjugated secondary antibody was incubated with the membrane for 1 hr at room temperature. The blot was detected by chemiluminescence (Clinx, Shanghai, China). The intensity of each band was quantified using ImageJ software (Media Cybernetics, Inc, Rockville, MD).

## Small interfering RNA transfection

The small interfering RNA (siRNA) was purchased from Tsingke Biotechnology Co., Ltd. (Beijing, China). The siRNA oligoconstruction used was as follows: sense 5′-GCACCAUGCAGAAUACAAATT-3′ and antisense 5′-UUUGUAUUCUGCAUGGUGCTT-3′. MC3T3-E1 cells were transfected using 50 nM siRNA mixed with Lipofectamine 3000 (Thermo Fisher Scientific, Inc, Waltham, MA) in a 12-well plate. All transfection procedures were performed in strict accordance with the Lipofectamine 3000 reagent protocol. The gene knockdown efficiency was confirmed by qRT-PCR and the rates of protein production were determined using western blotting.

## Animals and treatment

All of the animals were handled according to approved Institutional Animal Care and Use Committee (IACUC) protocols (Aproval No.: IACUC-20190318–03) of Zhejiang Chinese Medical University. All surgery was performed under sodium pentobarbital anesthesia, and every effort was made to minimize suffering. Ten-to-eleven-week-old Sprague-Dawley female rats (Vital River Laboratory Animal Technology Co., Ltd., China) were acclimated for 5 days before the experiments started. All rats were maintained under specific pathogen-free conditions at Zhejiang Chinese Medical University Laboratory Animal Research Center. Rats were either sham-operated (SHAM, n=7) or bilateral OVX (n=21) under general anesthesia. One week after recovering from surgery, the OVX rats were randomly divided into three groups: OVX with vehicle (OVX, n=7), OVX with low-dose 7,8-DHF (OVX-L, n=7, 5 mg/kg/day), and OVX with high-dose 7,8-DHF (OVX-H, n=7, 10 mg/kg/day). Masking was used during group allocation. 7,8-DHF was dissolved in 5% DMSO/95% methylcellulose (0.5%, wt/vol). Vehicle or 7,8-DHF solutions were all administrated intragastrically for 12 weeks. During the experimental period, all rats were allowed free access to distilled water and fed with standard rat chow.

The body weight of the rats was recorded weekly during the experimental period. Urine samples were collected from the rats that were housed individually for 24 hr in metabolic cages without providing food 1 day before the rats were sacrificed. Before laparotomy, blood samples were collected and serums were then prepared by centrifugation of the collected bloods. Urine and serum samples were both stored at −80°C for biochemical determinations. Uterine was removed from each rat and immediately weighed. Femurs were dissected and filled in 4% paraformaldehyde and stored at 4°C for measurement of BMD by DXA (Lunar Prodigy, GE Healthcare, Madison, WI), trabecular microarchitecture by microcomputed tomography (micro-CT, NEMO, PINGSENG Healthcare (Kunshan) Inc, Kunshan, China), and bone histomorphometry analysis. The right tibias were cleaned of adherent tissue, packed with physiological saline-soaked gauze and stored at −80°C for measurement of compression test.

## DXA analysis

Two-dimensional BMD of the left femur was measured using Lunar Prodigy by DXA (GE Healthcare, Madison, WI) equipped with appropriate software for bone density assessment in small laboratory animals. BMD was calculated by bone mineral content of the measured area.

## Micro-CT analysis

After DXA measurement, one representative left femur from each group was selected for evaluating trabecular microarchitecture of the femoral metaphysis using micro-CT (NEMO, PINGSENG Healthcare (Kunshan) Inc, Kunshan, China). The selection of representative sample was based on the

median value of BMD of respective group. The distal femur is rich in trabecular bone compared with the proximal and middle regions. Therefore, the femur was scanned from the proximal growth plate in the distal direction (50 μm/slice). This region included 3600 images obtained from each femur using 1024 × 1024 matrix resulting in an isotropic voxel resolution of 29 μm³. The volume of interest was selected as a region 30slices away from the growth plate at the proximal end of the femur to 70 slices. The 3D images were obtained for visualization and display.

### Compression test

The biomechanical properties of the fresh right tibia were assessed using a compression test on an electronic universal testing machine (Shenzhen SUNS Technology Stock Co., Ltd., Shenzhen, China). The test was performed using upper and lower jigs, vertical in the sagittal plane. The lower part of the tibia specimen was completely fixed in the lower jig using 502 glue. A compression load was applied at a rate of 2 mm/min until the tibia specimen fractured. During the test, the load-displacement data were obtained to calculate the biomechanical parameters, such as the maximum load, the fracture deflection, and the fracture strain.

### Bone histomorphometry

The slices of right femurs were subjected to hematoxylin and eosin (H&E) staining and TRAP staining. After being freed from soft tissues, the right femurs were fixed in 4% paraformaldehyde for 24 hr at 4℃ and/or decalcified and then dehydrated. Subsequently, the specimen was embedded in wax, made into 4 μm paraffin slices, and baked at 60℃. The slices were deparaffinated before being subjected to H&E staining and TRAP staining. After staining, the slices were sealed in neutral gum and analysis was performed with NIKON Eclipse Ci microscope equipped with a digital camera. TRAP-positive cells with three or more nuclei were identified as osteoclasts, and the osteoclast number/bone surface was calculated by Image-pro plus 6.0 (Media Cybernetics, Inc, Rockville, MD).

### Assay for serum and urine chemistry

S-Ca and ALP concentrations were determined using an automatic biochemical analyzer (Hitachi High-Technologies (Shanghai) Co., Ltd., Shanghai, China). U-Ca and creatinine (Cr) concentrations were analyzed by the same method used for the serum samples. Urinary excretion of Ca was expressed as the ratio to Cr concentration (Ca/Cr). In addition, serum levels of FSH, E2, and osteocalcin (BGP) were measured by ELISA kit (Cusabio Technology Llc., Wuhan, China) according to the manufacturer.

### Statistical analysis

Biological replicates are parallel measurements of biologically distinct samples and technical replicates are repeated measurements of the same sample (*Blainey et al., 2014*). The number of independent biological replicates was 3, unless otherwise stated. Other statistical details of the experiments could be found in the figure legends for each experiment, including the number of wells in the cell culture experiments and the number of rats in the animal experiments (represented as n, unless otherwise stated). Data were expressed as mean ± SD and analyzed using GraphPad Prism 9 software (GraphPad Software, Inc, La Jolla, CA). Statistical differences were assessed using one-way or two-way analysis of variance (ANOVA) was applied to assess the statistical significance. $p < 0.05$ was considered statistically significant.

## Acknowledgements

The authors are indebted to Dr Lizong Zhang and other staff at the Laboratory Animal Research Center, Zhejiang Chinese Medical University, for their help with animal experiments. We thank Prof. Yu Zhang for aiding in experimental equipment of qRT-PCR and western blotting. We are also grateful to all participants, including Dr Chun Chen, Dr Jing Xiong, Dr Yunhong Li, Dr Yufeng Chen, Mr Lu Jin, Mr Jinwei Lu, etc., for their enthusiastic participation in this study. This work was supported by the Key Research and Development Program of Guangdong Province Project (grant number: 2019B020212001).

## Additional information

### Funding

| Funder | Grant reference number | Author |
|---|---|---|
| Key Research and Development Program of Guangdong Province | 2019B020212001 | Ying Zhang |

The funders had no role in study design, data collection and interpretation, or the decision to submit the work for publication.

### Author contributions

Fan Xue, Resources, Data curation, Formal analysis, Validation, Investigation, Visualization, Writing - original draft; Zhenlei Zhao, Resources, Formal analysis, Investigation, Writing - original draft; Yanpei Gu, Formal analysis, Investigation, Visualization; Jianxin Han, Formal analysis, Visualization; Keqiang Ye, Conceptualization, Resources, Supervision, Project administration, Writing - review and editing; Ying Zhang, Conceptualization, Resources, Supervision, Funding acquisition, Project administration, Writing - review and editing

### Author ORCIDs

Fan Xue (iD) https://orcid.org/0000-0003-0057-3792
Ying Zhang (iD) https://orcid.org/0000-0001-8533-4138

### Ethics

Animal experimentation: All of the animals were handled according to approved Institutional Animal Care and Use Committee (IACUC) protocols (Approval No. IACUC-20190318-03) of Zhejiang Chinese Medical University. All surgery was performed under sodium pentobarbital anesthesia, and every effort was made to minimize suffering.

### Decision letter and Author response

Decision letter https://doi.org/10.7554/eLife.64872.sa1
Author response https://doi.org/10.7554/eLife.64872.sa2

## Additional files

### Supplementary files

• Supplementary file 1. Sequences of primers used for quantitative real-time PCR (qRT-PCR).

• Transparent reporting form

### Data availability

All data generated or analysed during this study are included in the manuscript and supporting files.

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
