## [Decision Letter]

**Acceptance summary:**

Your manuscript provides data on use of novel compounds that can mimic BDNF for potential clinical applications and targeting of specific tissues. This manuscript also provides additional data on BDNF effects on bone.

**Decision letter after peer review:**

Thank you for submitting your article "7,8-dihydroxyflavone Modulates Bone Formation and Resorption and Ameliorates Ovariectomy-Induced Osteoporosis" for consideration by *eLife*. Your article has been reviewed by 3 peer reviewers, including Carlos Isales as the Reviewing Editor and Reviewer #1, and the evaluation has been overseen by a Senior Editor.

The reviewers have discussed the reviews with one another and the Reviewing Editor has drafted this decision to help you prepare a revised submission.

As the editors have judged that your manuscript is of interest, but as described below that additional experiments are required before it is published, we would like to draw your attention to changes in our revision policy that we have made in response to COVID-19 (https://elifesciences.org/articles/57162). First, because many researchers have temporarily lost access to the labs, we will give authors as much time as they need to submit revised manuscripts. We are also offering, if you choose, to post the manuscript to bioRxiv (if it is not already there) along with this decision letter and a formal designation that the manuscript is "in revision at eLife". Please let us know if you would like to pursue this option. (If your work is more suitable for medRxiv, you will need to post the preprint yourself, as the mechanisms for us to do so are still in development.)

Summary:

The manuscript "7,8-dihydroxyflavone modulates bond formation and reabsorption and ameliorates ovariectomy-induced osteoporosis" by Xue et al. reports a novel role of BDNF mimetic, 7,8-dihydroxyflavone (7,8-DHF), on promoting osteoblast proliferation and differentiation but suppressing osteoclastogenesis via TrkB. The authors have also demonstrated the osteoporosis-preventive function of 7,8-DHF in ovariectomized rats, which might have significant translational values. While the study is logically designed and the findings further enhance our understanding of the connection between neurotrophin and bone remodeling, more detailed analyses are needed to fully delineate the molecular actions of 7,8-DHF in the skeletal system. The are several major concerns

Essential revisions:

1) In the introduction section, the author claimed that BDNF is a potential therapeutic target in numerous neurological, mental, and metabolic disorders. However, the poor delivery and short half-life of BDNF in vivo impose an insurmountable hurdle to its clinical application. The author should better explain the physiological role of BDNF in nervous system with a relative short half-life and the reason of the short half-life. Long half-life does not necessarily mean better. And why long-half-life BDNF like cytokine is needed in bone metabolism? What the molecular basis that 7,8-DHF can mimic BDNF? BDNF as a protein/peptide based functioning molecule, the function was deployed mainly through active peptide motif. 7,8-DHF however, as one of the flavonoids, is a chemical with two benzene rings with carbon connections. The two molecules are very different and how can one of them mimicking another? The author should provide molecular and structural evidence. The abstract is not appropriately written. It does even describe any results related to BDNF. The studies should include BDNF in the in vitro analyses because previous studies reported contradictorily that BDNF increased osteoclast formation (Sun et al. Int J Cancer 2012 130: 827; Ai et al., PLoS One 2012 7: e46287). Comparing the activity difference between endogenous ligand and non-peptidyl mimetics is essential to understand their functional specificity.

2) The authors used RAW264.7 and MC3T3-E1 immortal cell lines for studying in vitro effects and molecular clues. The changes are modest at best, use of primary cells is encouraged, and in vivo evidence needs to be strengthened.

3) The details of 7,8-DHF-provoked β-catenin/Run2 signaling has not been sufficiently addressed. How does 7,8-DHF increase the expression of β-catenin? Moreover, the subtle increase of β-catenin expression questions if it is the major downstream effector for inducing the expression of key osteoclastogenic factors Run2 and osterix. A β-catenin knockdown experiment in MC3T3-E1 cells should be performed to verify this conclusion. The possibility that 7,8-DHF enhances the activity of other Run2 inducers such as Smad2 and *TAK1* should not be excluded, especially that BDNF is able to induce BMP7 production (Ortega and Alcantara Cereb Cortex 2010 20: 2132) and enhance integrin expression in MC3T3-E1 (Zhang et al., J Cell Mol Med 2020 24: 10792).

4) In addition to osteoclastogenesis, the effect of 7,8-DHF on the reabsorption activity (e.g. Pit assay or OsteoAssay) of differentiated osteoclasts should be assessed. Expression of reabsorption markers such as Adamts5, Loxl2, etc should also be considered. This information will be important to evaluate if BDNF signaling is also important in modulating the activity of mature osteoclasts.

5) Explanations on the discrepant results between protein content and gene expression of Runx2, osterix, and β-catenin after 7,8-DHF stimulation should be provided (Figures 3E to 3G).

6) In Figure 5 some of the responses with the Low dose 7,8DHF are higher than Controls. What does 7,8 DHF do by itself, without OVX? A control experiment using 7,8 DHF without OVX would be helpful to interpret the response.

7) The n number (n=3) used to examine the effect of 7,8-DHF in osteoclast formation in vivo is not sufficient to give a convincing conclusion (Figure 6).

8) Statistics on the bone morphometric parameters should be provided (Table 1).

9) In Table 2, it appears that estradiol levels rise after OVX when using the 7,8 DHF, why is that?

---

## [Author Response]

Essential revisions:1) In the introduction section, the author claimed that BDNF is a potential therapeutic target in numerous neurological, mental, and metabolic disorders. However, the poor delivery and short half-life of BDNF in vivo impose an insurmountable hurdle to its clinical application. The author should better explain the physiological role of BDNF in nervous system with a relative short half-life and the reason of the short half-life.

BDNF plays a key role in the survival, differentiation, and development of neurons. It also influences oligodendrocyte proliferation and remyelination. In the central nervous system (CNS), neurons synthesize BDNF and it is suggested that BDNF mainly works in a paracrine or autocrine manner in the CNS (Makar, Nimmagadda, Trisler, and Bever, 2014). The result of a systemic application of BDNF in an experimental model of neuritis was negative, and the study found that BDNF had a short half-life and did not readily cross the brain blood barrier (BBB) (Felts, Smith, Gregson, and Hughes, 2002). The outcomes of several clinical trials using recombinant BDNF are disappointing, also possibly because of the short in vivo half-life and poor delivery of BDNF (Ochs et al., 2000; Thoenen and Sendtner, 2002).

BDNF exerts its biological functions on neurons predominantly via binding to the transmembrane receptor tyrosine kinase TrkB. BDNF triggers TrkB dimerization through conformational changes and autophosphorylation of tyrosine residues in the intracellular domain, resulting in activation of three major signaling pathways involving mitogen-activated protein kinase (MAPK), phosphatidylinositol 3-kinase (PI3K) and phospholipase C-γ (PLC-γ) (C. Liu, Chan, and Ye, 2016; Xia Liu et al., 2014). These signaling cascades have critical roles in neuronal plasticity, survival and neurogenesis (Z. Zhang et al., 2014). Nonetheless, BDNF-elicited TrkB signals are transient with a peak signal at 10 min and decreases at 60 min. Furthermore, BDNF provokes activated TrkB receptor to be ubiquitinated and degraded. Conversely, 7,8-DHF swiftly induces TrkB phosphorylation and the activation sustains for more than 3 h without inducing TrkB ubiquitination or degradation (C. Liu et al., 2016; Xia Liu et al., 2014).

Long half-life does not necessarily mean better. And why long-half-life BDNF like cytokine is needed in bone metabolism?

At present, antiosteoporosis drugs face many challenges, including short half-life. For example, salmon calcitonin (sCT), a polypeptide drug approved by the Food and Drug Administration (FDA), is widely used in more than 90 countries as a typical antiresorptive drug for the treatment of metabolic bone disease. However, the clinical application of sCT is limited by its short half-life because of its rapid clearance in vivo (Yu et al., 2019). NEL-like molecule-1 (NELL-1) protein, an osteogenic growth factor, is widely studied in bone regeneration. Similar to other native protein drugs, NELL-1 has a relatively short half-life time in vivo after intravenous injection, which limits its application as a therapy for osteoporosis. Due to the rapid clearance of native protein drug in vivo, high dose and frequent administration usually have to be adopted to achieve therapeutic benefit. This can lead to high treatment cost and low patient compliance in chronic treatment (Y. Zhang et al., 2014). Long half-life agents in the matrix are well positioned to support long-term gains in BMD, but also carry the risk of adverse skeletal effects (Estell and Rosen, 2021). For BDNF itself *per se*, it has the intrinsic problem as a polypeptide including poor PK profiles and too short half-life for clinical usage. 7,8-DHF obviously exhibits much better and improved PK features as compared to BDNF as a TrkB receptor agonist.

What the molecular basis that 7,8-DHF can mimic BDNF? BDNF as a protein/peptide based functioning molecule, the function was deployed mainly through active peptide motif. 7,8-DHF however, as one of the flavonoids, is a chemical with two benzene rings with carbon connections. The two molecules are very different and how can one of them mimicking another? The author should provide molecular and structural evidence.

BDNF triggers TrkB dimerization and autophosphorylation of tyrosine residues in its intracellular domain, resulting in activation of the downstream signaling pathways. Similarly，7,8-DHF specifically activate TrkB. It binds to the receptor extracellular domain of TrkB, promote the receptor dimerization and autophosphorylation, and activate the downstream signaling cascades (Jang et al., 2010). It is also a bioavailable chemical that can pass through the BBB to provoke TrkB and its downstream PI3K/Akt and MAPK activation. Besides, internalization of the neurotrophin-Trk complex plays a critical role in signal transduction that initiates cell body responses to target-derived neurotrophins. Both BDNF and 7,8-DHF strongly escalate TrkB internalization. The SAR (structural-activity relationship) study has shown that the 7,8-dihydroxy groups are essential for the agonistic effect, and 8-hydroxy group in A ring is essential for the TrkB stimulatory effect, as 5,7-DHF, 5,6-DHF and 5,6,7-THF could not activate TrkB. None of the single hydroxy flavone derivatives exhibited notable TrkB stimulatory effect and no trihydroxy flavones or dimethoxy flavones demonstrated any substantial effect (C. Liu et al., 2016; X. Liu et al., 2010).

The abstract is not appropriately written. It does even describe any results related to BDNF. The studies should include BDNF in the in vitro analyses because previous studies reported contradictorily that BDNF increased osteoclast formation (Sun et al., Int J Cancer 2012 130: 827; Ai et al., PLoS One 2012 7: e46287). Comparing the activity difference between endogenous ligand and non-peptidyl mimetics is essential to understand their functional specificity.

As requested, we have added the results of BDNF in the in vitro analyses to Figure 1A, 1D; Figure 4A-C; Figure 1—figure supplement 1, in addition to the Abstract. Unlike the above-mentioned research, which BDNF increased osteoclast formation, our study found that BDNF significantly inhibited the formation of TRAP-positive multinucleated osteoclasts. The differences are as follows: The cell model employed in our study was RAW 264.7 murine macrophages, a well-characterized cell linage with regard to osteoclastogenic study. However, Sun et al., and Ai et al., used human peripheral blood mononuclear cells and multiple myeloma cell lines. The concentrations of BDNF applied in our study were 25, 50, 100 ng/mL, while they used 0.1, 1, 5, 10, 25 ng/mL BDNF to determine the effect of BDNF on osteoclast formation and activity in vitro.

2) The authors used RAW264.7 and MC3T3-E1 immortal cell lines for studying in vitro effects and molecular clues. The changes are modest at best, use of primary cells is encouraged, and in vivo evidence needs to be strengthened.

MC3T3-E1 subclone 14, derived from mouse calvaria, is a good model for studying in vitro osteoblast differentiation, particularly extracellular matrix (ECM) signaling. They exhibit effects similar to primary calvarial osteoblasts (Wang et al., 1999). RAW264.7 murine cell line is an important tool for in vitro studies of OC formation and function, in parallel or as a prelude to studies with OCs formed from primary cells (Collin-Osdoby and Osdoby, 2012). The use of primary cells raised difficulties. Such limits including the availability and variation in response patterns among different cellular study preparations (Kong, Smith, and Hao, 2019). Despite all these hurdles, we appreciate the reviewer’s constructive comments and have conducted some experiments using primary cells. Human bone marrow-derived mesenchymal stem cells (BMSCs) and murine bone marrow macrophages (BMMs) were induced to osteoblasts and osteoclasts respectively as previously described (Hu et al., 2019; Zhang et al., 2018). The results were as follows (Author response image 1): 0.5 μM and 1 μM 7,8-DHF promotes osteogenic differentiation of BMSCs as ALP activity was increased significantly (*p<0.05*), though 7,8-DHF did not display an obvious effect on cell growth over the time range. Besides, 7,8-DHF had no cytotoxicity on survival of BMMs and possessed inhibitory effect on osteoclastic differentiation of BMMs, as the number of TRAP-positive multinucleated cells was sharply reduced by 7,8-DHF treatment (*p<0.001*).

**Author response image 1. respfig1:** 7,8-DHF promoted osteogenic differentiation of BMSCs and inhibited osteoclastic differentiation of BMMs. (A) The effect of 7,8-DHF on the proliferation of BMSCs after treatment for 24, 48 and 72 h. (B) The effect of 7,8-DHF on the ALP activity of BMSCs. Results were normalized with total protein quantity. (C) Representative images of TRAP-positive multinucleated osteoclasts after the treatment with 7,8-DHF for 5 d. (magnification: 100×, scale bar: 200 μm) (D)The average number of TRAP-positive multinucleated (nuclei ≥ 3) cells per cell. (D) The effects of 7,8-DHF on the cytoactive of BMMs. All results were expressed as mean ± SD. (A, E: n = 4; B-D: n = 3; **p < 0.05*, ***p < 0.01*, ****p < 0.001*, *****p < 0.0001*, ns: not significant, one-way ANOVA).

3) The details of 7,8-DHF-provoked β-catenin/Run2 signaling has not been sufficiently addressed. How does 7,8-DHF increase the expression of β-catenin?

The glycogen synthase kinase 3β (GSK3β) in the degradation complex is a key negative regulator of β-catenin, which phosphorylates β-catenin and further degrades it via the ubiquitin pathway in quiescent cells (Jain, Ghanghas, Rana, and Sanyal, 2017). As shown in Figure 2A and 2B, 7,8-DHF inactivated GSK3β by increasing its phosphorylation, leading to disrupting the β-catenin destruction complex. The p-GSK3β/GSK3β value significantly increased as the concentrations of 7,8-DHF elevated *(p<0.05)*.

Moreover, the subtle increase of β-catenin expression questions if it is the major downstream effector for inducing the expression of key osteoclastogenic factors Run2 and osterix. A β-catenin knockdown experiment in MC3T3-E1 cells should be performed to verify this conclusion. The possibility that 7,8-DHF enhances the activity of other Run2 inducers such as Smad2 and TAK1 should not be excluded, especially that BDNF is able to induce BMP7 production (Ortega and Alcantara Cereb Cortex 2010 20: 2132) and enhance integrin expression in MC3T3-E1 (Zhang et al., J Cell Mol Med 2020 24: 10792).

As requested, a β-catenin knockdown experiment in MC3T3-E1 cells was performed and results were shown in Figure 2J-M, Figure 1—figure supplement 1. Compared to negative control, expression of β-catenin, Runx2 and Osterix was markedly reduced by β-catenin knockdown in both the absence and presence of 7,8-DHF (*p<0.05*). β-catenin is the major downstream effector for inducing the expression of key osteoclastogenic factors Runx2 and osterix. Besides, the activities of other Runx2 inducers such as Smad2 and *TAK1* had no obvious difference with 7,8-DHF treatment as shown in Figure 2—figure supplement 2. Based on these results, we speculated that β-catenin is the major downstream effector for inducing the expression of key osteoclastogenic factors Runx2 and osterix. However, we did not carry out other experiments involving BMP7 or integrin β1, because the MAPK/ERK signaling pathway was not the main objective of this study, we primarily focused on investigating the specific signaling molecules involved in the Wnt/β-catenin pathway, in order to avoid complicating the issues.

4) In addition to osteoclastogenesis, the effect of 7,8-DHF on the reabsorption activity (e.g. Pit assay or OsteoAssay) of differentiated osteoclasts should be assessed. Expression of reabsorption markers such as Adamts5, Loxl2, etc should also be considered. This information will be important to evaluate if BDNF signaling is also important in modulating the activity of mature osteoclasts.

We appreciate the reviewer’s suggestion. Unfortunately, the Corning osteo assay surface multiple well plate (CLS3988) is not currently available, and it is hard to find suitable substitutes. Hence, we cannot perform the suggested Pit assay or OsteoAssay.

As requested, expression of Adamts5 was examined and results showed that 0.5 μM and 1 μM 7,8-DHF significantly decreased the protein level of Adamts5 (*p<0.05*) (Figure 4G). We did not measure the expression of Loxl2, because Loxl2 is expressed by pre-hypertrophic and hypertrophic chondrocytes in vivo, and that LOXL2 expression is regulated in vitro as a function of chondrocyte differentiation, not osteoclast differentiation (Iftikhar et al., 2011). Besides, Loxl2 promotes aggressive osteosarcoma, a malignant tumor of the bone whose pathological mechanisms is different from osteoporosis (Matsuoka et al., 2020). Instead, we determined another resorption marker matrix metalloproteinase-9 (MMP-9), an enzyme of the metalloproteinase family, which plays an important role in the bone matrix degradation and bone remodeling. As shown in Figure 4F, 0.5 μM and 1 μM, 7,8-DHF notably decreased the protein levels of MMP-9.

5) Explanations on the discrepant results between protein content and gene expression of Runx2, osterix, and β-catenin after 7,8-DHF stimulation should be provided (Figures 3E to 3G).

These results indicated that 7,8-DHF might increase the protein expression of β-catenin, Runx2 and osterix via a post-transcriptional regulation. In fact, it is not abnormal that the mRNA level of a gene is not consistent with the protein level. For instance, Gao et al. found that along with the curcumin treatment, the mRNA level of Krüppel-like factor 5 (KLF5) was not decreased significantly, which was not consistent with the protein level decrease (Gao et al., 2014). Khan et al. found that the mRNA expression pattern of β-galactosidase was not consistent with the protein level (Khan, Takahashi, Abe, and Komatsu, 2009).

6) In Figure 5 some of the responses with the Low dose 7,8DHF are higher than Controls. What does 7,8 DHF do by itself, without OVX? A control experiment using 7,8 DHF without OVX would be helpful to interpret the response.

We appreciate the reviewer’s constructive comments. In our previous research, we used a prodrug strategy for elevating 7,8-DHF oral bioavailability and the brain exposure, and found that the optimal prodrug R13 has favorable properties. R13 is readily hydrolyzed into 7,8-DHF in liver microsomes and the plasma concentration of 7,8-DHF is also significantly enhanced (Chen et al., 2018). Recently, we conducted an experiment using 7,8-DHF prodrug R13 to treat wild-type (WT) mice orally. Specifically, 12 weeks old WT mice received vehicle or R13 at dose of 21.8 mg/kg/d, six days per week, for 8 weeks by gavage. μCT scan and analysis was performed in femurs ex vivo using a μCT-40 scanner. The results were shown in Author response image 2. The structure and volume of trabecular bone, as well as the fretting and microdamage in cortical bone in WT+R13 group was improved compared with the WT+control group. The research, which includes the above results, is continuing and has yet to be published.

**Author response image 2. respfig2:** Representative samples were reconstructed in 3D to generate visual representations of trabecular and cortical structure.

7) The n number (n=3) used to examine the effect of 7,8-DHF in osteoclast formation in vivo is not sufficient to give a convincing conclusion (Figure 6).

We appreciate the reviewer’s suggestion and have made some modifications accordingly. As shown in Figure 6C, we increased the sample sizes to 5, ensuring that the parameter is more accurate and the results obtained are more persuasive.

8) Statistics on the bone morphometric parameters should be provided (Table 1).

We entrusted Pingsheng Healthcare (Kunshan) Inc (Jiangsu, China) to conduct the micro-CT analysis. The data Table 1 showed were raw data from the company, which did not have statistics. In order to make the results more reliable, we deleted the previous Table 1 and re-organized the manuscript.

9) In Table 2, it appears that estradiol levels rise after OVX when using the 7,8 DHF, why is that?

It was reported that follicle-stimulating hormone (FSH) promotes the differentiation of ovarian follicles and modulates estrogen secretion (Danilovich et al., 2000). We speculate that the rise of estradiol levels when using the 7,8-DHF is possibly due to reduction of FSH. Recently, our study showed that 7,8-DHF at the medium dose (10 mg/kg·BW) resulted in a marked increase in bone mineral density (BMD) irrespective of dietary conditions. All 7,8-DHF treated groups, whether fed a low-fat diet (LFD) or high-fat diet (HFD), had lower FSH levels and higher serum estradiol levels in female mice (Zhao et al., 2021). Based on what was emerging from this research, we included the results of serum FSH levels in bone biochemical indexes. As shown in Table 1, the increase of estradiol level was consistent with the decrease of FSH levels when using 7,8-DHF after OVX.

References

Chen, C., Wang, Z., Zhang, Z., Liu, X., Kang, S. S., Zhang, Y., and Ye, K. (2018). The prodrug of 7,8-dihydroxyflavone development and therapeutic efficacy for treating Alzheimer's disease. Proceedings of the National Academy of Sciences of the United States of America, 115(3), 578-583. doi:10.1073/pnas.1718683115

Collin-Osdoby, P., and Osdoby, P. (2012). RANKL-Mediated Osteoclast Formation from Murine RAW 264.7 cells. In M. H. Helfrich and S. H. Ralston (Eds.), Bone Research Protocols, Second Edition (Vol. 816, pp. 187-202).

Danilovich, N., Babu, P. S., Xing, W., Gerdes, M., Krishnamurthy, H., and Sairam, M. R. (2000). Estrogen Deficiency, Obesity, and Skeletal Abnormalities in Follicle-Stimulating Hormone Receptor Knockout (FORKO) Female Mice**This investigation was supported in part by the Canadian Institutes of Health Research. Endocrinology, 141(11), 4295-4308. doi:10.1210/endo.141.11.7765 %J Endocrinology

Estell, E. G., and Rosen, C. J. (2021). Emerging insights into the comparative effectiveness of anabolic therapies for osteoporosis. Nature Reviews Endocrinology, 17(1), 31-46. doi:10.1038/s41574-020-00426-5

Felts, P. A., Smith, K. J., Gregson, N. A., and Hughes, R. A. C. (2002). Brain-derived neurotrophic factor in experimental autoimmune neuritis. Journal of Neuroimmunology, 124(1-2), 62-69. doi:10.1016/s0165-5728(02)00017-6

Gao, Y., Shi, Q., Xu, S., Du, C., Liang, L., Wu, K.,... Guo, P. (2014). Curcumin Promotes KLF5 Proteasome Degradation through Downregulating YAP/TAZ in Bladder Cancer Cells. International Journal of Molecular Sciences, 15(9), 15173-15187. doi:10.3390/ijms150915173

Hu, B., Sun, X., Yang, Y., Ying, Z., Meng, J., Zhou, C.,... Yan, S. (2019). Tomatidine suppresses osteoclastogenesis and mitigates estrogen deficiency-induced bone mass loss by modulating TRAF6-mediated signaling. 33(2), 2574-2586. doi:https://doi.org/10.1096/fj.201800920R

Iftikhar, M., Hurtado, P., Bais, M. V., Wigner, N., Stephens, D. N., Gerstenfeld, L. C., and Trackman, P. C. (2011). Lysyl Oxidase-like-2 (LOXL2) Is a Major Isoform in Chondrocytes and Is Critically Required for Differentiation. Journal of Biological Chemistry, 286(2), 909-918. doi:10.1074/jbc.M110.155622

Jain, S., Ghanghas, P., Rana, C., and Sanyal, S. N. (2017). Role of GSK-3β in Regulation of Canonical Wnt/β-catenin Signaling and PI3-K/Akt Oncogenic Pathway in Colon Cancer. Cancer Investigation, 35(7), 473-483. doi:10.1080/07357907.2017.1337783

Khan, N. A., Takahashi, R., Abe, J., and Komatsu, S. (2009). Identification of cleistogamy-associated proteins in flower buds of near-isogenic lines of soybean by differential proteomic analysis. Peptides, 30(12), 2095-2102. doi:https://doi.org/10.1016/j.peptides.2009.08.012

Kong, L., Smith, W., and Hao, D. (2019). Overview of RAW264.7 for osteoclastogensis study: Phenotype and stimuli. Journal of Cellular and Molecular Medicine, 23(5), 3077-3087. doi:10.1111/jcmm.14277

Liu, C., Chan, C. B., and Ye, K. (2016). 7,8-dihydroxyflavone, a small molecular TrkB agonist, is useful for treating various BDNF-implicated human disorders. Translational Neurodegeneration, 5. doi:10.1186/s40035-015-0048-7

Liu, X., Chan, C. B., Jang, S. W., Pradoldej, S., Huang, J. J., He, K. Y.,... Ye, K. Q. (2010). A Synthetic 7,8-Dihydroxyflavone Derivative Promotes Neurogenesis and Exhibits Potent Antidepressant Effect. Journal of Medicinal Chemistry, 53(23), 8274-8286. doi:10.1021/jm101206p

Liu, X., Obianyo, O., Chan, C. B., Huang, J., Xue, S., Yang, J. J.,... Ye, K. (2014). Biochemical and Biophysical Investigation of the Brain-derived Neurotrophic Factor Mimetic 7,8-Dihydroxyflavone in the Binding and Activation of the TrkB Receptor. Journal of Biological Chemistry, 289(40), 27571-27584. doi:10.1074/jbc.M114.562561

Makar, T. K., Nimmagadda, V. K. C., Trisler, D., and Bever, C. T., Jr. (2014). Cell-Based Delivery of Brain-Derived Neurotrophic Factor in Experimental Allergic Encephalomyelitis. Journal of Interferon and Cytokine Research, 34(8), 641-647. doi:10.1089/jir.2013.0160

Matsuoka, K., Bakiri, L., Wolff, L. I., Linder, M., Mikels-Vigdal, A., Patino-Garcia, A.,... Wagner, E. F. (2020). Wnt signaling and Loxl2 promote aggressive osteosarcoma. Cell Research, 30(10), 885-901. doi:10.1038/s41422-020-0370-1

Ochs, G., Penn, R. D., York, M., Giess, R., Beck, M., Tonn, J.,... Toyka, K. V. (2000). A phase I/II trial of recombinant methionyl human brain derived neurotrophic factor administered by intrathecal infusion to patients with amyotrophic lateral sclerosis. Amyotrophic Lateral Sclerosis and Other Motor Neuron Disorders, 1(3), 201-206. doi:Doi 10.1080/14660820050515197

Thoenen, H., and Sendtner, M. (2002). Neurotrophins: from enthusiastic expectations through sobering experiences to rational therapeutic approaches. Nature Neuroscience, 5, 1046-1050. doi:10.1038/nn938

Wang, D., Christensen, K., Chawla, K., Xiao, G. Z., Krebsbach, P. H., and Franceschi, R. T. (1999). Isolation and characterization of MC3T3-E1 preosteoblast subclones with distinct in vitro and in vivo differentiation mineralization potential. Journal of Bone and Mineral Research, 14(6), 893-903. doi:10.1359/jbmr.1999.14.6.893

Yu, P., Chen, Y., Wang, Y., Liu, Y., Zhang, P., Guo, Q.,... Li, J. (2019). Pentapeptide-decorated silica nanoparticles loading salmon calcitonin for in vivo osteoporosis treatment with sustained hypocalcemic effect. Materials Today Chemistry, 14. doi:10.1016/j.mtchem.2019.08.008

Zhang, W., Chen, E., Chen, M., Ye, C., Qi, Y., Ding, Q.,... Pan, Z. (2018). IGFBP7 regulates the osteogenic differentiation of bone marrow-derived mesenchymal stem cells via Wnt/β-catenin signaling pathway. 32(4), 2280-2291. doi:https://doi.org/10.1096/fj.201700998RR

Zhang, Y., Velasco, O., Zhang, X., Ting, K., Soo, C., and Wu, B. M. (2014). Bioactivity and circulation time of PEGylated NELL-1 in mice and the potential for osteoporosis therapy. Biomaterials, 35(24), 6614-6621. doi:https://doi.org/10.1016/j.biomaterials.2014.04.061

Zhang, Z., Liu, X., Schroeder, J. P., Chan, C.-B., Song, M., Yu, S. P.,... Ye, K. (2014). 7,8-Dihydroxyflavone Prevents Synaptic Loss and Memory Deficits in a Mouse Model of Alzheimer's Disease. Neuropsychopharmacology, 39(3), 638-650. doi:10.1038/npp.2013.243

Zhao, Z., Xue, F., Gu, Y., Han, J., Jia, Y., Ye, K., and Zhang, Y. (2021). Crosstalk between the muscular estrogen receptor α and BDNF/TrkB signaling alleviates metabolic syndrome via 7,8-dihydroxyflavone in female mice. Molecular Metabolism, 45, 101149. doi:https://doi.org/10.1016/j.molmet.2020.101149